# Enhancing autophagy by redox regulation extends lifespan in *Drosophila*

Claudia Lennicke [1,2], Ivana Bjedov[3], Sebastian Grönke [4], Katja E. Menger [5], Andrew M. James [5], Jorge Iván Castillo-Quan [6], Lucie A. G. van Leeuwen[1,2], Andrea Foley[1,2], Marcela Buricova[1,2], Jennifer Adcott[6], Alex Montoya [1,2], Holger B. Kramer [1,2], Pavel V. Shliaha[1,2], Angela Logan[5], Filipe Cabreiro [2,7], Michael P. Murphy [5], Linda Partridge [4,6] ✉ & Helena M. Cochemé [1,2] ✉

Dysregulation of redox homeostasis is implicated in the ageing process and the pathology of age-related diseases. To study redox signalling by $H_2O_2$ in vivo, we established a redox-shifted model by manipulating levels of the $H_2O_2$-degrading enzyme catalase in *Drosophila*. Here we report that ubiquitous overexpression of catalase robustly extends lifespan in females. As anticipated, these flies are strongly resistant to a range of oxidative stress challenges, but interestingly are sensitive to starvation, which could not be explained by differences in levels of energy reserves. This led us to explore the contribution of autophagy, which is an important mechanism for organismal survival in response to starvation. We show that autophagy is essential for the increased lifespan by catalase upregulation, as the survival benefits are completely abolished upon global autophagy knock-down. Furthermore, using a specific redox-inactive knock-in mutant, we highlight the in vivo role of a key regulatory cysteine residue in Atg4a, which is required for the lifespan extension in our catalase model. Altogether, these findings confirm the redox regulation of autophagy in vivo as an important modulator of longevity.

Redox signalling is a fundamental process that impacts a diverse range of biological pathways[1–4]. Redox regulation can operate through the selective post-translational modification (PTM) of redox-sensitive cysteine residues in target proteins[1–4]. Cysteines are typically present in a reduced, thiol state in vivo (–SH)[5–8], but can undergo oxidation in response to specific stimuli, including levels of reactive oxygen species (ROS) or changes in the ratios of redox cofactors, such as NADPH/NADP+ and glutathione/glutathione disulfide. The nature of this oxidative PTM will vary according to the physiological context, but typically proceeds via a sulfenic acid (–SOH) intermediate[1–4].

The redox reactivity of a given cysteine will depend on a combination of physical and biochemical properties. For instance, the position of the cysteine within the tertiary protein structure will determine its accessibility to pro-oxidants, and ability to interact with other redox-dependent binding partners. Additionally, the cysteine $pK_a$ will be influenced by adjacent charged amino acid residues and the local cellular environment, which governs its protonation state and therefore susceptibility to oxidative PTM[1–4]. Critically, redox signalling is distinct from irreversible oxidative damage and functions as a reversible 'redox switch' to regulate target proteins. Dysregulation of redox homeostasis has long been implicated in the pathophysiology of many age-related diseases, as well as in the ageing process itself, however, the underlying mechanisms remain largely unclear[9,10].

$H_2O_2$ acts as the major effector of redox signalling, both directly and through intracellular thiol redox relays[11,12]. The oxidative PTMs induced by $H_2O_2$ lead to specific changes in cellular pathways,

[1]MRC Laboratory of Medical Sciences (LMS), London, UK. [2]Institute of Clinical Sciences, Imperial College London, Hammersmith Hospital Campus, London, UK. [3]UCL Cancer Institute, London, UK. [4]Max Planck Institute for Biology of Ageing, Cologne, Germany. [5]MRC Mitochondrial Biology Unit, University of Cambridge, Cambridge Biomedical Campus, Cambridge, UK. [6]Institute of Healthy Ageing and GEE, University College London, London, UK. [7]CECAD Research Cluster, University of Cologne, Cologne, Germany. ✉e-mail: linda.partridge@ucl.ac.uk; helena.cocheme@lms.mrc.ac.uk

including many processes linked with ageing[13,14]. To study redox signalling by $H_2O_2$ in vivo and explore its involvement in health and longevity, we used the fruit fly *Drosophila* as a model organism, with its tractable lifespan, powerful genetic tools, and strong evolutionary conservation of many central metabolic pathways with mammals[15]. Here we report that inducing an endogenous redox-shift, by manipulating levels of the $H_2O_2$-degrading enzyme catalase, improves health and robustly extends lifespan in female flies, independently of oxidative stress resistance and dietary restriction. We find that the catalase redox-shifted flies are sensitive to starvation, which relies on autophagy as a vital survival mechanism. Importantly, we show that autophagy is essential for the lifespan extension of the catalase upregulated flies, which is abolished in an autophagy-deficient background. Furthermore, using a redox-inactive knock-in mutant of Atg4a, a major effector of autophagy, we show that the lifespan extension in response to catalase requires a key redox-regulatory cysteine residue, Cys102 in *Drosophila* Atg4a. These findings demonstrate that redox regulation of autophagy can extend lifespan, confirming the importance of redox signalling in ageing and as a potential pro-longevity target.

## Results

### Catalase over-expression extends lifespan

To explore the role of endogenous redox signalling by $H_2O_2$ in vivo, we used the binary UAS/GAL4 expression system to upregulate catalase in wild-type (WT) flies. Global upregulation of catalase under control of the ubiquitous *daughterless* promoter (da-GAL4 > UAS-cat) extends the median and maximum lifespan of female flies (typically by ~10–15%; Fig. 1a and Supplementary Data 1). Importantly, for these experiments, we used the *white Dahomey* ($w^{Dah}$) background, which is a long-lived and outbred WT, hence, we are extending healthy lifespan and not rescuing a short-lived defect. Catalase was over-expressed ~5–10-fold at the mRNA level in whole flies (Supplementary Fig. 1a), which led to corresponding increases in catalase protein (Supplementary Fig. 1b, c) and enzymatic activity (Supplementary Fig. 1d). Interestingly, lifespan was not extended in males (Fig. 1a), despite similar catalase over-expression (Supplementary Fig. 1a, b, d). However, interventions such as modulating nutrient-sensing in *Drosophila* often show sex-specific effects on survival in females, that are absent or marginal in males[16]. Furthermore, over-expression of a mitochondria-targeted catalase (da-GAL4 > UAS-mito-cat) did not lead to lifespan extension in either females or males (Supplementary Fig. 1e, f), consistent with a previous study[17], suggesting that altered mitochondrial metabolism is not implicated.

The da-GAL4 > UAS-cat flies were mildly delayed in eclosing, without affecting the overall proportion of larvae surviving to adulthood (Supplementary Fig. 1g). To exclude developmental effects, we showed that the lifespan extension could be fully recapitulated using the inducible GeneSwitch system, with over-expression from d2 of adulthood onwards (da-GS > UAS-cat ± RU; Fig. 1b and Supplementary Fig. 1a). Varying the dose of the inducer drug RU (50–400 μM) still did not extend lifespan in males and had only marginal further effects on lifespan extension in females (Supplementary Fig. 1h), suggesting that the catalase benefits are already maximal. Over-expressing catalase under the control of an alternative ubiquitous driver (actin5c-GAL4 > UAS-cat) also extends lifespan in females (Supplementary Fig. 1i). Furthermore, catalase-mediated lifespan extension is independent of *Wolbachia* status (Supplementary Fig. 1j, k), which can influence fly longevity and physiology[18,19]. In addition to lifespan extension, the catalase females also exhibit increased healthspan[20], as inferred from their enhanced climbing ability with age (Fig. 1c). To explore effects on age-specific mortality, trajectories derived from the survival curves reveal a shift in the intercept, but not the slope (Fig. 1d), indicating that catalase upregulation decreased the overall risk of death, rather than slowing its rate of increase with age[21]. Therefore, the catalase over-expressing females are healthier for longer.

Using the inducible GeneSwitch system showed that induction of catalase from middle-age (d28 and d42) or old-age (d56) is sufficient to extend lifespan (Fig. 1e and Supplementary Fig. 1l), although not to the full extent as induction from d2. By d56, the −RU control flies have already started dying, yet switching to +RU treatment even at this late stage still enhances survival. This implies that for full benefits the redox shift needs to occur early in life, yet late-onset still offers protection. The level of catalase upregulation induced by RU is equivalent at all ages, as are the levels of endogenous catalase in the controls, eliminating any contribution from changes in RU consumption or endogenous catalase expression with age (Supplementary Fig. 1m). We conclude that ubiquitous upregulation of catalase improves healthspan and extends lifespan in female flies. Interestingly, tissue-specific catalase upregulation using a range of drivers (e.g., renal tubules, insulin-producing cells, pan-neuronal, intestine, fat body, pan-muscular; Supplementary Fig. 1n–s), did not recapitulate the strong lifespan extension obtained by the ubiquitous drivers, suggesting that catalase is either acting in an untested tissue (or combination of tissues), or alternatively is required at a global, organismal level. The catalase over-expressors are exceptionally resistant to multiple modes of oxidative stress−by exogenous $H_2O_2$ (Fig. 1f), the redox cycler paraquat both upon feeding (Fig. 1g) and injection (Fig. 1h), as well as hyperoxia (Fig. 1i and Supplementary Fig. 1t). However, this enhanced oxidative stress resistance is unlikely to explain the lifespan extension in females, because catalase over-expression protects males to a similar extent against oxidative stress without increasing longevity.

### Catalase-induced lifespan extension is not through dietary restriction (DR)

To explore the mechanism underlying the catalase-mediated lifespan extension, we examined its relationship to DR, which is a robust and evolutionary conserved nutritional intervention known to have health and longevity benefits[22]. We measured the lifespan response of catalase over-expressor females to DR by varying the yeast content (i.e., protein source) in the food, while maintaining the sugar content constant[23]. This generated a typical tent-shaped response (Fig. 2a), with lifespan decreased at very low yeast levels (0.1×), highest under restricted conditions (0.5×), then gradually shortened towards more fully-fed conditions (1.5×). The lifespan of the catalase flies is enhanced relative to controls at all yeast levels (Fig. 2a), while fecundity increases with yeast content throughout the 0.1–1.5× range for both the control and catalase females (Supplementary Fig. 2a). Therefore, the catalase over-expressor females exhibit a normal DR response, and the lifespan extension upon catalase upregulation is not mediated by the activation of DR pathways.

While the DR experiment reveal improved survival compared to control at a range of yeast concentrations (Fig. 2a), including extremely poor nutritional conditions (0.1×-yeast; Fig. 2b), we unexpectedly observed that the catalase over-expressing females, but not males, are more sensitive than control to complete starvation (Fig. 2c). There was no difference in triacylglyceride (TAG) levels both basally ($t = 0$) and during a starvation time course, either at d7 (Fig. 2d) or d28 (Supplementary Fig. 2b). Similarly, the levels of glycogen storage and mobilisation were the same in control and over-expressor females (Fig. 2e). Therefore, the starvation sensitivity of the catalase flies is not due to differences in metabolic energy reserves or their mobilisation.

### Autophagy is upregulated and required for the catalase-mediated lifespan extension

Autophagy is a known longevity assurance process, involved in the response to nutritional challenges such as starvation[24–28]. Furthermore, there is evidence for redox-regulation of autophagy[29]. Therefore, we next explored the involvement of autophagy in the differential starvation response and longevity of the catalase flies. To monitor

autophagy status, we stained fly midguts with LysoTracker Red, which labels acidic compartments such as late endosomes and lysosomes, including autophagolysosomes (Fig. 2f and Supplementary Fig. 2c). In parallel, we also monitored autophagic flux using the dye CytoID Green, which accumulates when autophagy is induced and blocked[30] (Supplementary Fig. 2d). The number of LysoTracker

punctae was significantly increased in the catalase over-expressors (Fig. 2f), without detectable CytoID staining (Supplementary Fig. 2e), indicating that autophagy is induced without blocking flux. Mitophagy levels were unchanged upon catalase upregulation, as inferred from the genetically-encoded mito-QC fluorescent reporter[31] (Supplementary Fig. 2f).

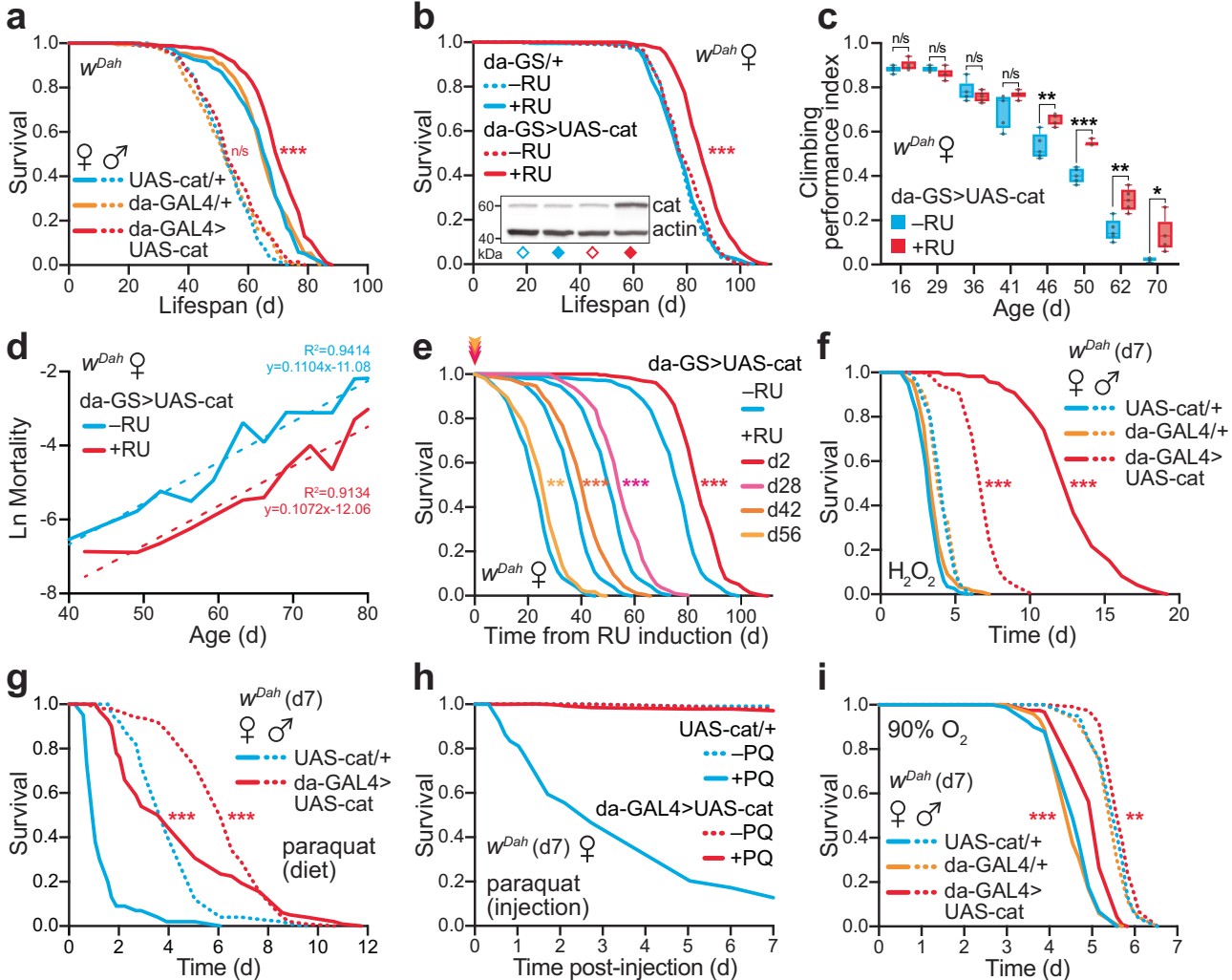

**Fig. 1 | Ubiquitous catalase upregulation extends lifespan in flies, independently of oxidative stress resistance. a** Constitutive, ubiquitous catalase over-expression (da-GAL4 > UAS-cat) extends the survival of female flies in a $w^{Dah}$ (*white Dahomey*) WT background relative to the UAS-cat/+ ($p = 5.7 \times 10^{-7}$) and da-GAL4/+ ($p = 3.3 \times 10^{-6}$) controls. The control lines are not significantly different from each other ($p = 0.7504$). No effect is observed in males ($p > 0.05$ for all comparisons). Lifespans were performed with $n = 200$ flies per condition. **b** Inducible catalase over-expression from early adulthood (d2) using the GeneSwitch system extends the lifespan of female flies (da-GS > UAS-cat ± RU, $p = 1.1 \times 10^{-16}$). RU has no effect on the da-GS/+ control line ($p = 0.7161$). Lifespans were performed with $n = 225–300$ flies per condition. Inset: catalase over-expression assessed by Western blotting in whole d9 females ( = d7 of RU induction), with actin as a loading control. **c** Healthspan, inferred from climbing performance, is improved in catalase over-expressing females. Climbing was assayed on da-GS > UAS-cat females ± RU to control for effects of eye colour on this behaviour. Data are presented as box-and-whisker plots (interquartile range, line at median, min/max error bars) of $n = 5$ replicates per condition, each with $n = 15$ flies per sample, analysed by unpaired two-tailed Student's *t*-test. **d** Mortality trajectories of the da-GS > UAS-cat ± RU survival curves from (**b**), fitted with a linear regression trendline (dotted line). **e** Late onset over-expression of catalase using the inducible GeneSwitch system from either middle-age (d28 and d42) and old-age (d56) extends the lifespan of female

flies ($p = 7.2 \times 10^{-8}$, $p = 1.0 \times 10^{-7}$ and $p = 1.4 \times 10^{-3}$, respectively against the −RU control). Lifespans were performed with $n = 270$ flies per condition, and were plotted from point of RU induction relative to the remaining −RU control flies at that age (see Supplementary Fig. 1l for the full survival data). **f** Catalase over-expressing flies are strongly resistant to exogenous $H_2O_2$ stress relative to controls (da-GAL4 > UAS-cat *v.* UAS-cat/+; $p = 5.0 \times 10^{-61}$ females, $p = 3.2 \times 10^{-33}$ males). $H_2O_2$ treatment (5% v/v in sucrose/agar medium) was initiated at d7, with $n = 105$ males ($n = 75$ for da-GS > UAS-cat) and $n = 120$ females per condition. **g** Catalase over-expressing flies are resistant to chronic dietary paraquat stress relative to control flies (da-GAL4 > UAS-cat *v.* UAS-cat/+; $p = 6.93 \times 10^{-31}$ females, $p = 7.99 \times 10^{-16}$ males). Paraquat treatment (20 mM in SYA food) was initiated at d7, with $n = 100$ flies per condition. **h** Catalase over-expressing flies (da-GAL4 > UAS-cat) are resistant to acute paraquat stress relative to controls (UAS-cat/+). d7 females were injected with 75 nL of 1 mg/mL paraquat in Ringers buffer (+ PQ, $n = 150$ flies) or mock injected with buffer alone (−PQ, $n = 120$ flies). **i** Catalase over-expressing flies are resistant to environmental hyperoxia stress relative to controls (da-GAL4 > UAS-cat *v.* UAS-cat/+; $p = 1.7 \times 10^{-8}$ females, $p = 8.1 \times 10^{-3}$ males). Incubation at 90% $O_2$ was initiated at d7, with $n = 120$ flies per condition (except $n = 90$ for UAS-cat/+ females). All survival assays (**a**, **b**, **e**, **f**, **g**, **i**) were analysed by Log-Rank test (see Supplementary Data 1 for full $n$ numbers and $p$ values). n/s, $p > 0.05$; *, $p < 0.05$; **, $p < 0.01$; ***, $p < 0.001$. Source data are provided as a Source Data file.

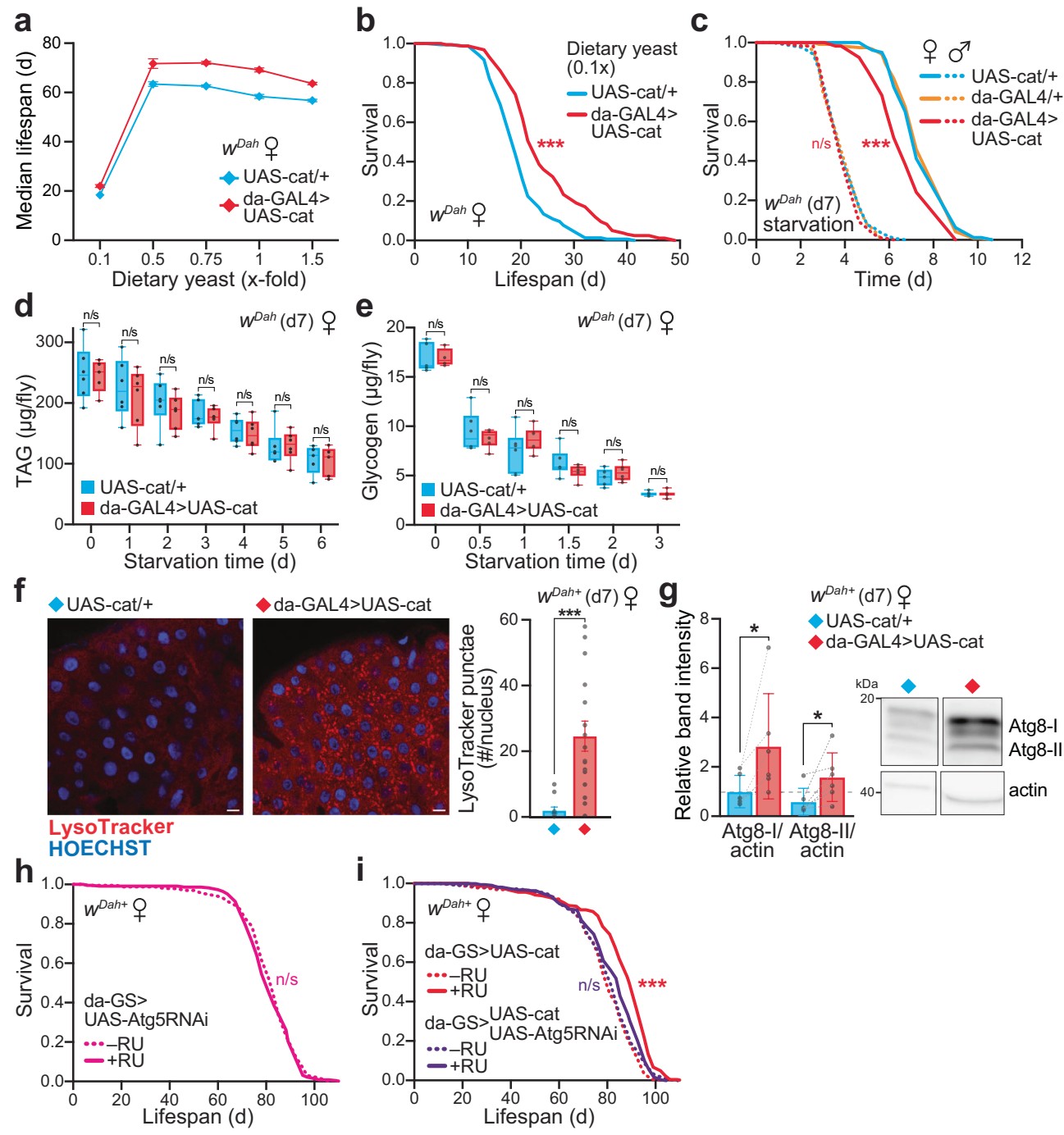

To assess autophagy by an orthologous approach, we quantified the levels of Atg8 (LC3 in mammals), a major autophagosome marker, by immunoblotting (Fig. 2g and Supplementary Fig. 2g). The levels of both the unlipidated (Atg8-I) and lipidated (Atg8-II) forms were strongly elevated in the catalase over-expressors, confirming that autophagy is induced. Indeed, levels of autophagy are physiologically fine-tuned, with both autophagy inhibition and excessive activation shown to induce starvation sensitivity in vivo[32–34]. Furthermore, similarly to the starvation stress assay, the catalase flies are also sensitive to treatment with the autophagy inhibitor chloroquine (Supplementary Fig. 2h).

To test the involvement of autophagy in the longevity of the catalase flies, we downregulated autophagy by RNAi of Atg5 (Supplementary Fig. 2i), which impairs autophagy[32,34]. Atg5 knock-down enhances sensitivity to starvation (Supplementary Fig. 2j), but does

not affect the lifespan of control females under fed conditions (da-GS > UAS-Atg5RNAi ± RU; Fig. 2h), consistent with previous reports[34]. Importantly, the lifespan extension by catalase over-expression is abolished in an Atg5-RNAi background (da-GS > UAS-Atg5RNAi+UAS-cat ± RU; Fig. 2i). Therefore, autophagy is required for the enhanced longevity by catalase upregulation.

## Redox proteomics of the catalase upregulated flies reveals an oxidising shift in bulk thiols

Redox regulation of autophagy has been described for Atg4 in the context of starvation-induced ROS production in vitro[35]. Atg4 is the only cysteine peptidase amongst the autophagy components, and is essential for autophagosome biogenesis[29]. Atg4 regulates autophagy by processing Atg8 at two critical stages: (1) the initial cleavage of Atg8, mediated by the redox-insensitive catalytic cysteine of Atg4, therefore

**Fig. 2 | Catalase-mediated lifespan extension requires autophagy. a** Catalase over-expressor (da-GAL4 > UAS-cat) and control (UAS-cat/+) females display a normal response to dietary restriction (DR). Median lifespan is plotted against the yeast content of the diet, with 1× corresponding to standard SYA food. Data are the means ± range of $n = 2$ independent lifespan experiments, each set up with $n = 150–160$ flies per genotype. **b** Survival curve on 0.1×-fold yeast from (**a**). The catalase over-expressor females (da-GAL4 > UAS-cat) are longer-lived than controls (UAS-cat/+) under low yeast nutritional conditions ($p = 3.6 \times 10^{-10}$, $n = 160$ flies per genotype). **c** Catalase over-expressor females (da-GAL4 > UAS-cat) are sensitive to starvation stress relative to UAS-cat/+ and da-GAL4/+ controls ($p = 5.6 \times 10^{-5}$ and $p = 7.6 \times 10^{-7}$, respectively). No difference is observed in males ($p > 0.05$ for all comparisons). Assays were performed at d7 with $n = 120$ flies per condition (except $n = 80$ for UAS-cat/+ females). **d, e** Triacylglyceride (TAG, **d**) and glycogen (**e**) levels in whole females assayed at d7 ($t = 0$), and depletion in response to starvation treatment. Data are presented as box-and-whisker plots (interquartile range, line at median, min/max error bars) of $n = 4–6$ replicates per genotype, each with $n = 5$ females per sample, analysed by unpaired two-tailed Student's $t$-test ($p > 0.05$). **f, g** Catalase over-expressor females (da-GAL4 > UAS-cat) display enhanced autophagy induction compared to UAS-cat/+ controls at d7. **f** LysoTracker Red staining of d7 female midguts quantified as the number of punctae relative to HOECHST-stained nuclei (scale bar = 10 μm). Data are means ± SEM of $n = 12$ (UAS-cat/+) and $n = 16$ (da-GAL4 > UAS-cat) biological replicates, analysed by unpaired two-tailed Student's $t$-test ($p = 2.6 \times 10^{-4}$). **g** Western blotting against Atg8, normalised to actin. Data are means ± SD of $n = 6$ biological replicates, each with $n = 10$ abdomens per sample, analysed by paired two-tailed Student's $t$-test (Atg8-I, $p = 0.0476$; Atg8-II, $p = 0.414$). Right, typical bands probed against Atg8 with actin as a loading control (see Supplementary Fig. 4e for the full blot). **h** Global Atg5 knock-down does not decrease lifespan in a WT background under control conditions (da-GS > UAS-Atg5RNAi ± RU; $p = 0.4177$). Lifespan assays were performed on $n = 225–240$ females. **i** Lifespan extension upon catalase over-expression (da-GS > UAS-cat ± RU; $p = 7.2 \times 10^{-15}$) is abolished in an autophagy-deficient background (da-GS > UAS-cat+UAS-Atg5RNAi ± RU; $p = 0.1701$). Survival assays (**b, c, h, i**) were analysed by Log-Rank test (see Supplementary Data 1 for full $n$ numbers and $p$ values). n/s, $p > 0.05$; *, $p < 0.05$; ***, $p < 0.001$. Source data are provided as a Source Data file.

this first step promoting Atg8 lipidation is redox-independent; and (2) the subsequent redox-dependent de-lipidation of Atg8, which is selectively inactivated upon oxidation of an adjacent redox-regulatory cysteine in Atg4. Under oxidising conditions, lipidated Atg8 accumulates due to the redox-driven suppression of de-conjugation by Atg4, therefore enhancing autophagosome biogenesis and promoting Atg4-mediated autophagy (Fig. 3a).

We previously showed that fasting for 24 h is associated with a strong oxidising shift of bulk cysteine residues in vivo in *Drosophila*[6]. We therefore hypothesised that the starvation sensitivity of the da-GAL4 > UAS-cat females may be attributed to such thiol redox changes. To explore the effects of catalase upregulation on global thiol redox state, we applied the same redox proteomic technique (oxidative isotope-coded affinity tags, OxICAT)[6] to the catalase over-expressing females. In OxICAT, samples undergo differential labelling of cysteine residues according to their redox status, followed by trypsin proteolysis and enrichment for cysteine-containing peptides, and finally detection by tandem mass spectrometry (Supplementary Fig. 3a). This allows both the identification of redox-responsive cysteine residues, as well as determination of their redox state. The bulk redox state of cysteines in control flies does not change with age, with the majority remaining at ~10–15% oxidised[6]. In contrast, the catalase over-expressors displayed an oxidising shift in cysteine redox state relative to controls with increasing age (Fig. 3b–d, Supplementary Fig. 3b–e and Supplementary Datas 2 and 3). This finding is surprising, since we are over-expressing an antioxidant enzyme. Indeed, we confirmed that global ROS levels are decreased upon catalase upregulation, as inferred from the redox-sensitive fluorescent dye CellROX (Supplementary Fig. 3f, g). Mitochondrial $H_2O_2$ levels, measured using the in vivo mitochondria-targeted ratiometric mass spectrometry probe MitoB[36], are unchanged in the da-GAL4 > UAS-cat females compared to UAS-cat/+ controls (Supplementary Fig. 3h), reflecting the fact that this catalase transgene is not mitochondria-targeted.

To reconcile catalase upregulation with the oxidising shift in bulk cysteine redox state, we tested the hypothesis that by quenching cytoplasmic $H_2O_2$, catalase over-expression blocks physiological $H_2O_2$-mediated redox signalling that upregulates other antioxidant systems and redox couples. The Keap1/Nrf2 signalling pathway is an appealing candidate for this process, as it is an oxidative stress response pathway that enhances the expression of multiple redox systems and is known to be redox-regulated in *Drosophila*[37]. To assess this pathway, we used a transgenic reporter for Keap1/Nrf2 activity, gstD-GFP. This pathway was upregulated with age in controls, but not in the long-lived catalase flies (Fig. 3e), suggesting that catalase over-expression prevented the induction of Keap1/Nrf2 signalling with age, and thus a range of downstream redox processes. Consistent with this finding, total levels of glutathione—an important cellular antioxidant and redox cofactor, whose synthesis is upregulated by Nrf2 signalling—are decreased in the catalase over-expressors (Fig. 3f). Altogether, we have shown that catalase flies undergo an unexpected global oxidising thiol redox shift with age. This oxidation is consistent with the enhancement of autophagy via redox-regulation of Atg4.

## Cys102 in Atg4a is required for the redox-mediated lifespan extension

The protein sequence of Atg4a is evolutionarily conserved, with both the catalytic cysteine (Cys98 in *Drosophila*) and the adjacent redox-regulatory cysteine (Cys102 in *Drosophila*) present in flies and mammals. This preservation is highlighted by structural modelling of *Drosophila* Atg4a and alignment analysis against human ATG4A (Fig. 4a, b and Supplementary Fig. 4a). To dissect the physiological role of Atg4a redox-regulation in vivo, we generated a transgenic knock-in fly line by CRISPR, where the regulatory cysteine in endogenous Atg4a is replaced by a redox-inactive serine residue (C102S mutant). This C102S knock-in mutation does not affect Atg4a expression (Supplementary Fig. 4b), or levels of catalase activity both basally and upon genetic upregulation (Supplementary Fig. 4c). Catalase over-expression in Atg4a-C102S knock-in flies displays the same strong resistance to $H_2O_2$-induced oxidative stress as in the Atg4a-WT background (Supplementary Fig. 4d).

Basal levels of autophagy are not affected under control conditions in the redox knock-in line (UAS-cat/+, Atg4a-WT v. UAS-cat/+, Atg4a-C102S), whereas autophagy induction by catalase over-expression is abolished (da-GAL4 > UAS-cat, Atg4a-C102S), as assessed both by Western blotting against Atg8 (Fig. 4c and Supplementary Fig. 4e) and LysoTracker staining (Fig. 4d and Supplementary Fig. 4f). Therefore, this redox-regulatory cysteine in Atg4a is required for autophagy induction by redox signalling in vivo, as previously reported in vitro[35].

To interrogate the role of Atg4a-Cys102 in mediating the longevity of the catalase flies, we performed survival assays with the Atg4a-WT CRISPR control line, and reproduced the catalase lifespan extension in this background (Fig. 4e). The Atg4a-C102S point mutation does not affect survival of control flies, confirming that this knock-in alone is not deleterious (Fig. 4f). Critically, in contrast to the Atg4a-WT control, the lifespan extension upon catalase upregulation is fully abolished in the Atg4a-C102S mutant background (Fig. 4f), as well as the climbing benefits (Supplementary Fig. 4g). While catalase upregulation leads to decreased egg laying (Supplementary Fig. 4h), this is observed in both the Atg4a-WT and Atg4a-C102S backgrounds, therefore any changes in fecundity cannot explain the lifespan extension. Overall, redox-regulation of autophagy via Atg4a-Cys102 mediates the longevity

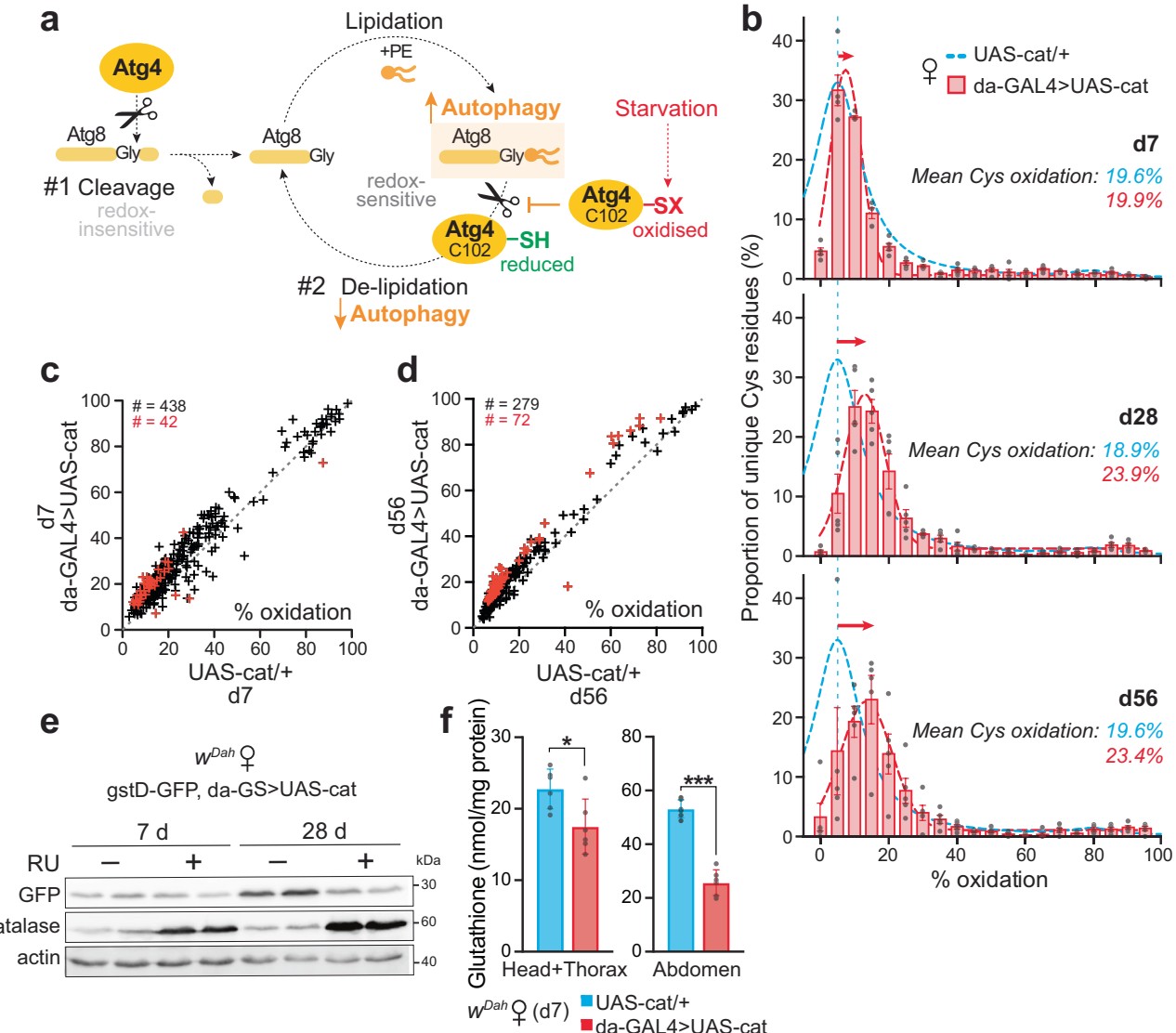

**Fig. 3 | Catalase flies undergo an oxidising shift in global thiol redox state.**
**a** Scheme showing the dual function of Atg4 in autophagy: (1) initial redox-independent cleavage of Atg8, to expose a C-terminal glycine residue enabling lipidation by PE (phosphatidyl-ethanolamine) via the E1-like enzyme Atg7 and the E2-like enzyme Atg3. The conjugated Atg8-PE is involved in autophagosome elongation/closure; (2) redox-dependent de-lipidation of Atg8-PE, allowing interaction and fusion of the autophagosome with the endosomal-lysosomal compartments, and recycling of cleaved Atg8. Oxidation of a redox-regulatory cysteine (C102 in *Drosophila* Atg4a) selectively inactivates the Atg8-PE de-conjugation activity of Atg4, promoting autophagosome biogenesis and therefore enhancing Atg4-mediated autophagy. **b** Redox proteomic (OxICAT) analysis of d7, d28 and d56 catalase over-expressing females (da-GAL4 > UAS-cat) compared to control (UAS-cat/+). Distribution of total cysteine residue oxidation levels, plotted as the proportion of the total number of peptides containing unique cysteine residues in each 5% quantile of percentage oxidation. Data are means ± SEM of $n = 5$ biological

replicates. **c**, **d** Oxidation state of cysteine residues present, comparing control versus catalase over-expressor females at d7 (**c**) and d56 (**d**). Data points above the diagonal dotted line (slope = 1) indicate cysteine residues more oxidised upon catalase upregulation, with red symbols designating significance ($p < 0.05$), assessed by unpaired two-tailed Student's *t*-test. The total number of unique Cys-containing peptides is indicated in black. **e** Levels of Nrf2 signalling inferred using a gstD-GFP reporter in da-GS > UAS-cat females treated ± RU for d7 and d28. Western blotting against GFP, catalase and actin, showing $n = 2$ biological replicates per condition. **f** Levels of total glutathione measured in d7 control (UAS-cat/+) and catalase over-expressor (da-GAL4 > UAS-cat) females. Data are means ± SEM of $n = 6$ biological replicates, each with $n = 10$ females per sample, analysed by unpaired two-tailed Student's *t*-test (head+thorax, $p = 0.0216$; abdomens, $p = 4.5 \times 10^{-7}$). n/s, $p > 0.05$; *, $p > 0.01$; ***, $p > 0.001$. Source data are provided as a Source Data file.

upon catalase over-expression. Enhancing autophagy is an evolutionarily conserved intervention associated with health and survival benefits, and here we demonstrate that selective redox-mediated upregulation of autophagy can extend lifespan.

## Discussion

Many attempts have been made to extend lifespan in model organisms by enhancing their antioxidant capacity, notably through the over-expression of antioxidant enzymes, including catalase[38–41]. These trials

were largely unsuccessful, casting doubt on the causative role of ROS and oxidative damage in ageing[42]. Therefore, our finding that catalase over-expression extends lifespan was at first surprising. The original study over-expressing catalase in *Drosophila* found no effect on lifespan and only modest resistance to oxidative stress by $H_2O_2$[38]. However, the study used an extra chromosomal copy under its endogenous promoter, resulting in lower over-expression of catalase (~1.75-fold at the mRNA level and ~1.5-fold increased enzyme activity). The degree of catalase over-expression is therefore likely to be important for the

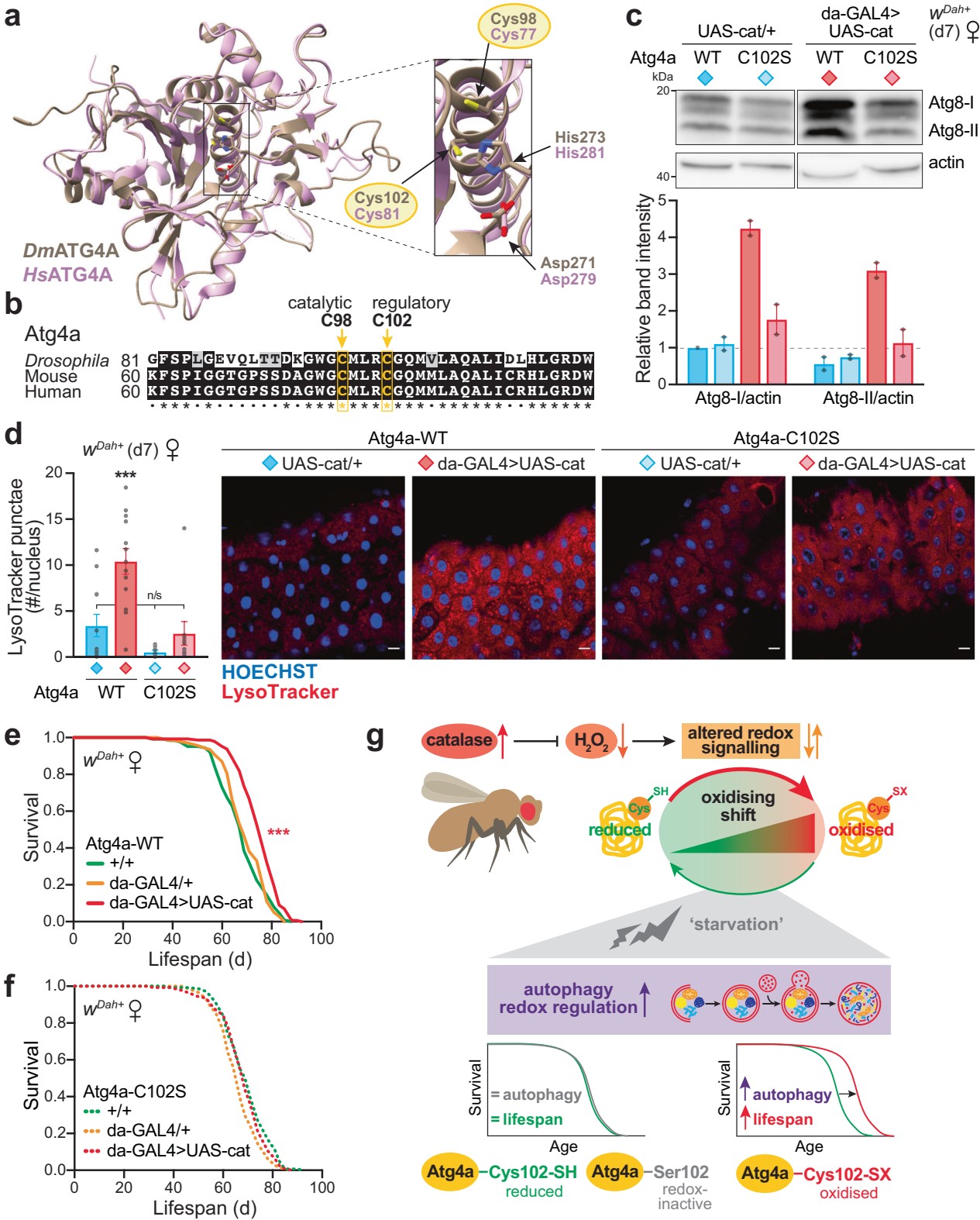

**e** *w*^Dah+ ♀ — Atg4a-WT
**f** *w*^Dah+ ♀ — Atg4a-C102S

lifespan extension. Furthermore, the earlier study used only males, while our findings show a robust effect specifically in females. The results of these studies are therefore reconcilable.

The sex-specificity of lifespan benefits upon catalase upregulation is striking. Indeed, we find that ubiquitous catalase over-expression does not extend lifespan or induce starvation sensitivity in males. A recent study showed that basal autophagy levels are significantly higher in male flies[43], shedding light on this sex-specificity of the

catalase effect. This suggests that females have a broader 'metabolic plasticity', and are more likely to benefit from interventions that enhance autophagy.

Importantly, our study also challenges widespread assumptions regarding the directionality of antioxidant interventions. Indeed, we find that in vivoupregulation of the antioxidant catalase, while directly conferring strong resistance to acute oxidative stress challenges as expected, also unexpectedly leads to an indirect oxidising shift in bulk cysteine

**Fig. 4 | Redox regulation of autophagy via Atg4a Cys102 extends lifespan. a** Structural representation of *Drosophila* Atg4a (brown, modelled with Phyre2[73]), superimposed on the crystal structure of human ATG4A (pink, PDB 2P82). Conservation of the catalytic triad and the regulatory cysteine (Cys102/81 in *Drosophila* and human, respectively) are highlighted. Structures were visualised in Chimera X[74]. **b** Multiple sequence alignment of the Atg4a protein from *Drosophila* with the mouse and human orthologues, showing the catalytic cysteine residue (Cys98 in *Drosophila*) and the redox-regulatory cysteine (Cys102 in *Drosophila*). See Supplementary Fig. 4a for the full sequence. **c, d** Autophagy induction in response to catalase over-expression is maintained in d7 Atg4a-WT female flies, but abolished in the Atg4a-C102S mutant background. **c** Autophagy levels assessed by Western blotting against Atg8 (see Supplementary Fig. 4e for the full blot). Quantification of Atg8 levels by densitometry, normalised to actin as a loading control. Data are means ± range of $n = 2$ biological replicates. **d** LysoTracker Red staining of midguts quantified as the number of punctae relative to HOECHST-stained nuclei (scale bar = 10 μm). Data are means ± SEM of $n = 8$–13 biological replicates, analysed by one-way ANOVA (Tukey). UAS-cat/+ *v.* da-GAL4 > UAS-cat comparison: Atg4a-WT, $p = 7.1 \times 10^{-4}$; Atg4a-C102S, $p = 0.731$. **e, f** Catalase over-expression extends lifespan in the Atg4a-WT control (**e**), but not in the redox-insensitive Atg4a-C102S knock-in background (**f**). +/+ control *v.* da-GAL4 > UAS-cat comparison: Atg4a-WT, $p = 1.8 \times 10^{-8}$; Atg4a-C102S, $p = 0.2154$. Survival assays (**e, f**) were analysed by Log-Rank test (see Supplementary Data 1 for full *n* numbers and *p* values). **g** Graphical summary: Catalase upregulation leads to the quenching of $H_2O_2$ as a second messenger molecule, which results in altered physiological redox signalling. Redox-responsive pathways, such as Nrf2, are not induced, causing an oxiding shift in bulk cysteine thiol redox state, which mimics an internal state of starvation. Autophagy is induced via redox-regulation of Cys102 in Atg4a as a protective response mechanism, which provides health and survival benefits. n/s, $p > 0.05$; ***, $p < 0.001$. Source data are provided as a Source Data file.

redox state by interfering with homeostatic redox signalling. Consistent with our observations, a recent study in *C. elegans* has shown that excess dietary supplementation with *N*-acetyl cysteine (NAC) accelerates ageing by inhibiting *skn-1*-mediated transcription, the worm orthologue of Nrf2[44]. Therefore, caution is required in the interpretation of experiments involving the genetic or dietary manipulation of antioxidants.

Our unbiased redox proteomic analysis has revealed that the catalase flies undergo an oxidising shift in bulk cysteine redox state with age. Interestingly, this pattern is similar to our earlier observations in control flies under starvation[6]. Nutrient deprivation can directly affect redox homeostasis by depleting the provision of important building blocks for reducing equivalents, such as NADPH and glutathione, mediating an intracellular oxidising shift[45]. Therefore, we suggest that the catalase-induced thiol oxidising shift is perceived as an internal state of starvation, which triggers the induction of autophagy as a protective response (Fig. 4g). Autophagy plays a fundamental role in healthy physiology, such as cellular differentiation, tissue remodelling, and mitochondrial homeostasis, as well as in the response to stress and the clearance of cellular damage[27]. Consequently, enhanced autophagy is a common denominator of many evolutionary conserved interventions that extend lifespan[26,27], both genetically, such as downregulation of insulin signalling[46], and pharmacologically, for instance rapamycin treatment[47]. Furthermore, direct upregulation of autophagy has been shown to exert health benefits and extend lifespan in a range of model organisms including worms, flies and mice[34,48–50]. Several components of the autophagy pathway are known to be redox-regulated, including Atg3 and Atg7[51], as well as the focus of our study Atg4[35,52].

Overall, we have shown that shifting the in vivo redox state of *Drosophila* by over-expression of catalase extends lifespan and healthspan in females through redox-regulation of autophagy via a key redox-responsive cysteine in Atg4a. Our findings further emphasise the importance of fine-tuning autophagy in health and disease, and demonstrate how manipulation of redox signalling in vivo can ameliorate the effects of ageing. Furthermore, our data are consistent with a growing view in the ageing field that many effects of ROS on longevity are likely to be through alterations in redox signalling rather than oxidative damage[9,10,53].

## Methods

### Fly strains and husbandry
The *white Dahomey* ($w^{Dah}$) strain of *Drosophila melanogaster* was used as the WT background. The *Dahomey* stock was collected in 1970 in Dahomey (presently the Republic of Benin), and maintained since then as large population cages, ensuring outbreeding and overlapping generations. The $w^{Dah}$ stock was derived by incorporation of the $w^{1118}$ mutation into the outbred *Dahomey* background by back-crossing. Flies were either negative ($w^{Dah}$) or positive ($w^{Dah+}$) for the bacterial cytoplasmic endosymbiont *Wolbachia*, with infection status confirmed by PCR using published primers against *wsp*[18]. The $w^{Dah}$ stock was originally achieved by tetracycline treatment of the $w^{Dah+}$ stock[18]. The following transgenic lines were used: UAS-cat (BDSC 24621), UAS-catRNAi (VDRC 6283), gstD-GFP[37], UAS-Atg5RNAi[32], UAS-mito-cat[54], UAS-mito-QC[31]. The following driver lines were used: da-GAL4[55], da-GS[56], act5c-GAL4 (BDSC 4414), Uro-GAL4[57], dilp2-GAL4[58], elav-GS (BDSC 43642), $S_1$106-GS (BDSC 8151), TIGS-2[59], and MHC-GS[60]. All transgenic lines were back-crossed into the appropriate $w^{Dah}$ or $w^{Dah+}$ background for at least 6–10 generations. Experimental flies were incubated at 25 °C on a 12 h light:12 h dark cycle with 65% humidity.

### Generation of the Atg4a lines
The Atg4a-C102S mutant fly line was generated by a fully transgenic CRISPR/Cas9 approach using two guide RNAs (gRNAs)[61] targeting the Atg4a (CG4428) gene. The C102S mutation together with an amino-terminal FLAG-tag were introduced using a donor construct for homology directed repair. As a control, a fly line carrying only the FLAG-tag, referred to as Atg4a-WT, was generated using the same approach. The gRNA construct was generated by PCR with Phusion polymerase (NEB) and primers SOL897 and SOL898 (see Supplementary Table 1 for primer sequences) and the plasmid pCDF4 as template[61]. The resulting PCR product was cloned into the pCDF4 vector using the Gibson Assembly kit (NEB) and the resulting plasmid was then used to generate ATG4-gRNA transgenic flies via injection into the $y^1$, $sc^1$, $v^1$, *P{y[+t7.7]=nos-phiC31\int.NLS}X; P{y[+t7.7]=CaryP} attP2* fly line (BDSC 25710). Atg4a donor constructs were cloned in a two-step approach. First, a 2792 bp part encompassing the Atg4a gene locus was amplified by PCR using SOL926 and SOL927 on BAC clone CH321-39B1 (BACPAC Resources) as a template and cloned into the pBluescript II KS vector via *NotI* and *KpnI* restriction mediated ligation. In the next step, gene synthesis (Eurofins Genomics) was used to synthesise 806 bp long fragments encoding an amino-terminal FLAG-tag, the C102S mutation and mutations within the PAM sites to introduce the Atg4a-C102S mutation, or an amino-terminal FLAG-tag and mutations within the PAM sites to generate the Atg4a-WT construct, respectively. These fragments were then used to replace the wild-type sequence in the 2792 bp construct using the endogenous restriction sites *BstBI* and *MfeI*. In order to introduce the mutation, Atg4a-gRNA transgenic flies were crossed with flies expressing Cas9 under the ubiquitous actin promotor, and their progeny was injected with the Atg4a-C102S and Atg4a-WT donor constructs. PCR screening with primers SOL955 and SOL929 that specifically target the FLAG-tag sequence was used to identify positive CRISPR events and the presence of the C102S mutation was confirmed by sequencing. All injections were performed by the transgenic fly facility of the Max Planck Institute for Biology of Ageing.

### Fly media
Flies were raised on standard sugar-yeast-agar medium (SYA) consisting of: 5% w/v sucrose (granulated sugar, Silver Spoon), 10% w/v

brewer's yeast (#903312, MP Biomedicals), 1.5% w/v agar (A7002, Sigma), supplemented with nipagin (H5501, Sigma; 30 mL/L of 10% w/v nipagin in 95% EtOH) and propionic acid (P1386, Sigma; 0.3% v/v) as mould inhibitors, added once the food had cooled down to ~60 °C[23]. Expression via the inducible GeneSwitch system was achieved by addition of the drug RU (RU486/mifepristone; M8048, Sigma) to standard SYA once cooled down to ~60 °C, typically at 200 μM from a 0.1 M stock in EtOH. For dietary restriction (DR) experiments, the yeast content was varied to give 1% (0.1×), 5% (0.5×), 7.5% (0.75×), 10% (1× = SYA) or 15% (1.5 SYA) w/v yeast[62].

## Experimental flies
For all experiments, eggs were collected over a defined period (< 24 h) to ensure a synchronous population and reared at constant density in 200 mL bottles with SYA[62]. Eclosing adults of a defined age were kept as a mixed population for ~48 h to allow mating, then separated into males and females under mild $CO_2$ anaesthesia, and maintained as separate sexes from then onwards.

## Lifespan & stress assays
Lifespan assays were set up as above, typically with $n$ ~ 10–15 flies per vial and a total of n ~ 100–250 flies per condition. Flies were transferred to fresh food without gassing every ~2–3 days, with deaths and censors recorded. Stress assays were performed on d7 flies (typically $n > 100$ per condition in groups of ~15–20 flies per vial), with deaths scored regularly following initiation of treatment. See Supplementary Data 1 for full survival assay information. For $H_2O_2$ resistance, flies were transferred onto medium containing 5% v/v $H_2O_2$ (H1009, Sigma), 5% w/v sucrose, 1.5% w/v agar. For paraquat stress, flies were either transferred onto standard SYA food supplemented with 20 mM paraquat (856177, Sigma), or injected with 75 nL of 1 mg/mL paraquat in Ringers buffer (3 mM $CaCl_2$, 182 mM KCl, 46 mM NaCl, 10 mM Tris base, pH 7.2 HCl) and maintained on standard SYA[63]. Starvation stress was assayed by transferring flies to 1.5% w/v agar medium, which lacks nutrients but allows hydration. Chloroquine (C6628, Sigma) was prepared as 10 mM in SYA or 5% w/v sugar, 1.5% w/v agar. Hyperoxia was performed by incubating flies on standard SYA vials in a glove box chamber set at 90% $O_2$ using a ProOx controller (BioSpherix). The majority of lifespans (Figs. 1a, b, e, 2a, b, h, I and 4e, f and Supplementary Fig. 1j–l) and stress assays (Figs. 1f, g, 2c, 4d and Supplementary Figs. 2h, j and 4d) were repeated at least twice as independent biological experiments, except Fig. 1h,i and Supplementary Fig. 1e, f, h, I, n–s which were performed once.

## Development time
Eggs were collected from flies in cages onto grape juice agar plates over a defined time window (~ 4 h). After ~24 h, the resulting L1 larvae were picked onto SYA food at a density of 50 per vial ($n = 500$ total per genotype), and the time to adult eclosion was monitored.

## Climbing assay
Climbing ability (negative geotaxis) was assayed essentially as described[63]. Briefly, groups of 15 flies were transferred to a sawn-open 25 mL serological pipette (35 cm long, 1.5 cm diameter), with the base sealed by parafilm. The flies were tapped down within the column and observed during 45 s, after which their location was recorded. The column was separated into three sections: top 10 cm, middle, bottom 3 cm. Each cohort was evaluated 3 times, using 5 groups per genotype. The climbing performance index was calculated as: $1/2 (n^{total} + n^{top} - n^{bottom}/n^{total})$.

## Metabolic and molecular assays
Flies for molecular experiments were rapidly transferred to pre-chilled microtubes via a small plastic funnel and snap frozen in liquid nitrogen, then stored at −80 °C until required. Flies were always frozen at approximately the same time of day to minimise any circadian variation. For some assays, frozen flies were separated into body segments using forceful manual impact[64].

## Quantitative PCR (QPCR)
Total RNA was extracted using Trizol (Ambion) according to the manufacturer's instructions (typically $n = 10$ flies per sample, with $n = 4$–5 replicates per condition). For some experiments, RNA was extracted from head and thorax tissue (i.e., abdomens removed), to avoid interference from the ovaries as the UAS/GAL4 system does not express in the germline. The concentration of total RNA purified for each sample was measured spectrophotometrically. 3 μg of total RNA was then subjected to DNA digestion using DNAse I (Ambion), immediately followed by reverse transcription using the Superscript II system (Invitrogen) with oligo dT primers. QPCR was performed using Fast SYBR Green Master Mix (Applied Biosystems) in a 7900HT Fast Real-Time PCR machine, and results processed by SDS2.4 software (Applied Biosystems), or using SYBR Green Supermix (1725124, Biorad) analysed with the QuantStudio 7 Flex Real-Time PCR System (Thermo Scientific). The primers were designed using Primer BLAST[65] and are listed in Supplementary Table 2.

## Western blotting
Frozen fly samples were homogenised directly into 2× Laemmli loading buffer (Bio-Rad) supplemented with 5% v/v β-mercaptoethanol (Sigma) using a pellet pestle and motor (usually 5/10 females into 100/200 μL) and separated by standard SDS-PAGE. The following primary antibodies were used at the indicated dilutions: anti-actin (Ab1801, Ab8224, or Ab8227, AbCam; 1:1000), anti-Atg8 (a generous gift from K. Köhler[66]; 1:1000), anti-catalase (C0979, Sigma; 1:10,000), anti-GFP (#2955, Cell Signaling Technology; 1:1000). The following secondary antibodies were used: anti-mouse IgG (#7076, Cell Signaling Technology, or A4416, Sigma; 1:5000), anti-rabbit IgG (#7074, Cell Signaling Technology; 1:5000). Blots were developed using standard ECL, followed by analysis with FIJI software[67].

## Catalase activity
Catalase activity was measured according to a published protocol[68], adapted for *Drosophila* samples. Briefly, 5 whole flies were homogenised in 200 μL PBS supplemented with protease inhibitor (HALT 1861284, ThermoScientific) using a TissueLyser (Precellys, 30 s), and centrifuged (16,000 × $g$, 4 °C, 10 min). 50 μL supernatants (1:10 diluted in PBS for the da-GAL4 > UAS-cat genotype) were transferred to flow cytometry tubes (Falcon 5 mL polystyrene, round bottom, 12 × 75 mm), and combined with 50 μL 1% Triton-DX100, followed by 50 μL 30% $H_2O_2$ solution (Sigma H1009). After incubating at RT for 10 min, the height of the stable foam was marked, the tubes photographed and analysed using FIJI. Enzyme activity was calculated by linear regression against a standard curve of purified catalase (C40, Sigma) dissolved in PBS.

## Glutathione assay
Total glutathione levels were measured as described[69], adapted for *Drosophila* samples. Briefly, cohorts of 10 female flies (separated into head+thorax and abdomen fractions[64]) were homogenised in 200 μL assay buffer (0.1 M potassium phosphate (pH 7.5), 5 mM EDTA), supplemented with 0.1% v/v Triton X-100 and 0.6% sulfosalicylic acid. Samples were centrifuged (16,000 × $g$, 4 °C, 10 min), and diluted 1:10 in assay buffer. 20 μL samples and glutathione standards (G6529, Sigma) were transferred to a 96-well plate and overlayed with 120 μL of assay buffer containing DTNB (D8130, Sigma; 2 mg/ 3 mL) and 10 U glutathione reductase (G3664, Sigma; 10 U/3 mL), followed by the addition of 60 μL NADPH (N7505, Sigma; 2 mg/3 mL assay buffer). The formation of TNB was monitored kinetically at 412 nm over 5 min in a spectrophotometer (Fluostar Omega, BMG Labtech).

Glutathione levels were determined from the slope relative to the standard curve.

## Energy storage assays

Whole body triacylglyceride (TAG) and glycogen levels were measured in d7 and d28 females ($n = 5$ flies per sample, $n = 6$–12 replicates per genotype) under control (fed) conditions and in response to starvation. For the TAG assay, flies were homogenised in 0.05% v/v Tween-20 and assayed using the Triglyceride Infinity Reagent (TR22421, ThermoScientific) in a 96-well plate measuring absorbance at 540 nm. For the glycogen assay, d7 flies were homogenised in saturated sodium sulphate, then the subsequent pellet was resuspended in anthrone reagent (319899, Sigma) and assayed in a 96-well plate measuring absorbance at 620 nm[70].

## OxICAT sample preparation and LC-MS/MS analysis

To measure the bulk redox state of protein cysteine residues, we performed unbiased redox proteomics using OxICAT. Protein isolation, cysteine-residue labelling, peptide preparation and LC-MS/MS analysis were performed exactly as described previously[6]. Briefly, cohorts of $n = 10$ females flies were rapidly frozen in liquid $N_2$, and the frozen head and thorax tissue was separated from the abdomens[64] to avoid signal from the ovaries. A total of $n = 5$ biological replicates were processed per condition. Samples were homogenised in ice-cold 100% w/v TCA to stabilise thiols and solubilise proteins. Homogenates were centrifuged to pellet the chitin exoskeleton and other insoluble components, and the resulting supernatant transferred to a fresh tube. Protein samples were precipitated by decreasing the TCA concentration to 20% w/v with addition of $H_2O$, then washed successively with 10% and 5% w/v TCA. This protein precipitation was shown to occur with minimal protein losses or distortion of the protein complement[6].

Protein samples (~30–40 µg) were resuspended in denaturing alkylating buffer (DAB; 6 M urea, 2% w/v SDS, 200 mM Tris-HCl, 10 mM EDTA, 100 µM DTPA, 10 µM neocuprine). Reduced Cys residues were labelled with light ICAT reagent (4339036, AB Sciex) for 2 h at 37 °C, 1400 rpm in an Eppendorf Thermomixer. Proteins were precipitated and washed with ice-cold acetone, solubilised in DAB with 1 mM TCEP (tris(2-carboxyethyl)phosphine) to reduce previously reversibly oxidised Cys residues, which were then labelled with heavy ICAT reagent (4339036, AB Sciex) as above, before again precipitating and washing in ice-cold acetone. ICAT-labelled protein samples were resuspended in DAB and digested with trypsin overnight at 37 °C. Digested samples were enriched for Cys-containing peptides first on a cation exchange cartridge, and then subsequently on an avidin affinity cartridge (both provided with the ICAT kit; 4339036, AB Sciex). The eluted peptides were dried down overnight in a SpeedVac, and the biotin moiety of the ICAT label cleaved.

Liquid chromatography-tandem mass spectrometry (LC-MS/MS) analysis of the OxICAT-labelled peptides was performed exactly as published[6], using an Orbitrap LTQ XL (Thermo) after chromatography on a nanoscale reverse-phase column. Raw files for each LC-MS/MS run were analysed using MaxQuant software to determine the ratio of heavy over light OxICAT-labelled peptides. Each biological sample ($n = 5$ per condition) was run as 2 technical replicates, with the raw files from both LC-MS/MS runs grouped into a combined dataset. Besides light or heavy ICAT labelling of Cys residues, methionine oxidation was included as a possible modification, and up to 2 possible missed cleavages were allowed in the MaxQuant search parameters[6]. As a reference sequence database, a FASTA file containing all protein sequences associated with *D. melanogaster* was generated from UniProt, and used to identify peptides. In addition, the reference sequence database was used to create an in silico tryptic digest of the *D. melanogaster* proteome, listing all Cys-containing peptides (with 0, 1 or 2 missed cleavages), the Cys residue number, and the UniProt protein ID[6]. Ratios of heavy over light ICAT-labelled peptides obtained from

MaxQuant were converted to % of the Cys residue reversibly oxidised (combining intensities from all peptide signals containing the Cys residue of interest (e.g., miscleaved, methionine oxidation, different z values), generating a mean % reversible oxidation for each unique Cys residue identified within the biological replicates (Supplementary Data 2). Functional annotations were obtained from DAVID[71] (https://david.ncifcrf.gov/tools.jsp; Supplementary Data 3).

## Confocal microscopy

Autophagy and autophagic flux were assessed by LysoTracker Red and CytoID Green staining, respectively[30]. Briefly, intact guts were dissected into 1× PBS (70011-036, Gibco) and stained in the dark for 30 min with CytoID Green (Autophagy Detection Kit 2.0, ENZ-KIT175, Enzo Life Sciences; 1:500 in 1× PBS), followed by a second incubation for 3 min in PBS with LysoTracker Red (L7528, Invitrogen; 1:2000 in 1× PBS) and HOECHST (H1399, Invitrogen; 1:1000 from a 1 mg/mL stock in ultrapure water) followed by 3 washes with PBS. Guts were mounted in mounting buffer (50% glycerol in PBS) and imaged immediately. As a positive control for CytoID staining, d2 female flies were fed for 7 days on standard SYA medium supplemented with 200 µM rapamycin (R-5000, LC Laboratories), then 48 h on a diet containing 200 µM rapamycin and 10 mM chloroquine prior to gut dissection and staining as described above. The number of LysoTracker Red and CytoID Green punctae relative to HOECHST-stained nuclei was quantified using CellProfiler software[72] (www.cellprofiler.org).

Cellular ROS levels were assessed using CellROX Deep Red (C10422, Invitrogen; 1:500 in PBS). Guts were dissected with minimal light exposure, and stained for 30 min at 25 °C in the dark, then washed 3 times with 1× PBS prior to mounting (50% glycerol in PBS, with HOECHST 1:1000 from a 1 mg/mL stock). Mean fluorescence intensity (MFI) was analysed using FIJI. As a positive control, dissected guts were treated ex vivo with 10 µM $H_2O_2$ for 10 min in 1× PBS at RT prior to CellROX staining as described above.

For catalase staining, whole guts were dissected in PBS and fixed in 4% paraformaldehyde (30450002-2, 2B Scientific Ltd) for 20 min. After 3 × 5 min washes with PBST (0.1% v/v Triton X-100 in PBS), guts were blocked with 5% w/v BSA (A7906, Sigma) in PBST for 1 h. Guts were incubated overnight at 4 °C with a primary anti-catalase antibody (C0979, Sigma; 1:250 in 5% w/v BSA in PBS). After three washes with PBS, guts were incubated with an anti-mouse secondary antibody (A10037, Invitrogen; Alexa568) for 1 h at RT, washed and mounted in VectaShield containing DAPI (H1200-10, Vector Laboratories). Images were acquired on a Leica SP5 confocal microscope (zoom: 20× optical plus 5× digital; resolution: 512 × 512 px at 700 Hz). Laser power and optical settings were kept constant between images.

## Structural modelling

The structure of *Drosophila* ATG4a was modelled using Phyre2[73] and visualised in Chimera X[74], superimposed on the crystal structure of human ATG4A (PDB 2P82). Distances between the amino acids belonging to the catalytic triad, as calculated in Chimera X[74], were comparable between human and *Drosophila*:

Human: His281_NE2↔SG_Cys77 3.286 Å, His281_ND1↔OD1_Asp279 2.616 Å;

*Drosophila*: Hs273_NE2↔SG_Cys98 3.413 Å, His273_ND1↔OD1_Asp271 2.620 Å

## MitoB injections

To assay in vivo mitochondrial $H_2O_2$ levels, d7 females were injected with the ratiometric mass spectrometry probe MitoB under mild $CO_2$ anaesthesia as described[36,64]. Flies were then returned to standard food (cohorts of 10 flies, $n = 6$–7 replicates per condition) and incubated for 4 h on SYA medium at 25 °C, before snap freezing. A $t = 0$ h control ($n = 3$ replicates) was snap frozen immediately to determine the background level of MitoP. Flies were homogenised, spiked with

deuterated internal standards, and extracted to quantify the MitoP/MitoB ratio[36,64].

## Statistical analysis

Lifespan and stress assays were plotted as cumulative survival curves, and statistical analysis was performed by Log-Rank test. Full details of survival data ($n$ numbers, $p$ values) are provided in Supplementary Data 1. Other data were analysed by Student's $t$-test or ANOVA as appropriate in GraphPad Prism v8–10.

## Reporting summary

Further information on research design is available in the Nature Portfolio Reporting Summary linked to this article.

## Data availability

The proteomics dataset has been deposited to the ProteomeXchange Consortium[75] via the PRIDE partner repository (accession number #PXD060330, with the control condition corresponding to #PXD002195 as published previously[6]). All other data supporting the findings from this study are available within the manuscript and its Supplementary Information, or from the corresponding authors upon reasonable request. Source data are provided with this paper.

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

## Acknowledgements

This project has received funding from the Medical Research Council UK (MC-A654-5QB90) to H.M.C., and from the Max Planck Society to L.P. M.P.M. is supported by the Medical Research Council UK (MC_UU_00028/4) and by a Wellcome Trust Investigator Award (220257/Z/20/Z). F.C. was supported by the Wellcome Trust/Royal Society (Sir Henry Dale Fellowship 102532/Z/12/Z and 102531/Z/13/A) and the DFG, German Research Foundation (EXC 2030-390661388). We are grateful to Pirrko Salmiheimo, Mumtaz Ahmad and Mary O'Sullivan for help with preparation of fly media, Jigna Patel for early technical support, Mike Harbour and Ian Fearnley for assistance with redox proteomics, and Yu-Xuan Lu and Sara Salgueiro Torres for advice on confocal imaging. We thank members of the Partridge and Cochemé labs for valuable discussions throughout this project. We acknowledge the Bloomington *Drosophila* Stock Center and the Vienna *Drosophila* Resource Center (VRDC) for fly strains, and the BACPAC Resource Center for BAC clones. For the purpose of open access, the author has applied a Creative Commons Attribution (CC BY) licence to any Author Accepted Manuscript version arising.

## Author contributions

Conceived the project: L.P. and H.M.C. Performed experiments and data analysis: C.L., I.B., J.I.C.Q., L.A.g.v.L., A.F., M.B., J.A., A.M., H.B.K., P.V.S., A.L., F.C. and H.M.C. Generated the CRISPR mutants: S.G. Conducted and analysed the redox proteomics: K.E.M. and A.M.J. Supervised the work: M.P.M., L.P. and H.M.C. Wrote the manuscript: C.L., M.P.M., L.P. and H.M.C., with input from all the authors.

## Competing interests

The authors declare no competing interests.
