## [Transparent Peer Review file · Nature Communications]

Enhancing autophagy by redox regulation extends lifespan in *Drosophila*

Corresponding Author: Professor Linda Partridge

Version 0:

Reviewer comments:

Reviewer #1

(Remarks to the Author)

Cocheme and coauthors have shown an importance of redox regulation in relation to longevity and stress resistance. Authors also showed that this mechanism works by affecting oxidation of specific residues in atg proteins. The work might be of great interest for scientific community but still some questions remained to make solid conclusions. They are as follow:

I wonder why authors have chosen 5S-10Y diet for most experiments? Fruit fly require more sugar for successful survival and high protein diet shortens the lifespan. Taken into account that yeasts have about 45% of protein and 24% carbs than P:C ratio of experimental diet is about 1:2 that is rather to maximize reproduction than lifespan. Data from Lee et al. (2009) show that P:C 1:8 is more supportive for maximal lifespan. So, the presented data was generated at rather protein-enriched diet that is presumable toxic for flies.

I couldn't find an information whether genetic manipulations affected an amount of food consumption? It is also reasonable in terms of determination of resistance to oxidative stress under H₂O₂ or paraquat. Latter one strongly decreases food palatability. So, the question is did all flies consumed same amount of oxidants? Moreover, did consumption of hydrogen peroxide and paraquat induced oxidative stress???

Authors have overexpressed catalase and showed it increased amount by WB. I wonder if they have measured enzymatic activity of catalase. WB may represent an increase in protein but it is not necessary that it is active. Antibodies can also bind inactive apoenzyme.

As far as I know hydrogen peroxide itself cannot cause oxidation of cysteine residues. It may be due to its single electron reduction to generate hydroxyl radical that is very reactive. Moreover, residues can be oxidized by superoxide anion that is a substrate for SOD. I suggest to overexpress SOD and check some most affected parameters. What about other enzymes that detoxify hydrogen peroxide – peroxidases or transferases?

On the supplementary fig 1B authors showed that cat overexpression slows down the development. Might be shorter development a causative for extended lifespan showed on fig 1A?

There is also no reasonable explanation why cat overexpression did not extend the lifespan in male flies. Is presented mechanism works only in females?? Additionally, as shown on supplementary fig2A, fecundity was decreased in cat overexpressing flies that can be a reason for lifespan modulation in female flies.

I also wonder if diet supplementation with antioxidant containing thiol groups will give the same phenotype. Cysteine, glutathione or DTT can be tested.

Reviewer #2

(Remarks to the Author)

In this manuscript, Cochemé and colleagues explored the role of redox signaling in the regulation of aging and metabolic

health, using *Drosophila* as a model organism. More specifically, the authors show that overexpression of catalase, driven by ubiquitous promoters early in adulthood, is required to enhance healthspan and longevity in female flies. Notably, both female and male flies that overexpress catalase are resistant to multiple modes of oxidative stress. In an effort to gain insight into the mechanism underlying the longevity-promoting effects of catalase overexpression, the authors found that catalase over-expressing female flies exhibited a normal response to dietary restriction (DR). However, these flies displayed an increased starvation sensitivity compared to controls despite the fact that the levels of triacylglyceride (TAG) and glycogen storage did not differ significantly between control and catalase overexpressing flies. Furthermore, catalase overexpressing flies had increased levels of autophagy and their lifespan extension was abolished by knockdown of Atg5. Implementation of the oxidative isotope-coded affinity tags (OxICAT) approach revealed that catalase overexpressing female flies displayed an oxidizing shift in cysteine redox state relative to controls during aging. This shift was dependent on the redox sensitive cysteine residue Cy102 in Atg4, the only cysteine peptidase in the autophagic pathway. The authors mutated Atg4 α -C102S by CRISPR-Cas9 and found that this mutation is sufficient to suppress lifespan extension conferred by catalase overexpression. The study provides evidence suggesting that ubiquitous overexpression of catalase (an enzyme that reduces hydrogen peroxide to water) extends lifespan in female flies. This finding is surprising since it has been previously shown that ectopic expression of catalase in the mitochondrion has either neutral or negative effects on longevity in *Drosophila* (Mockett et al., 2010, *Free Radical Biology & Medicine*; Sanz, 2016, *Biochimica et Biophysica Acta*). In line with these findings, it has been recently shown that catalase overexpression abrogates the extended lifespan associated with mild muscle mitochondrial perturbations in *Drosophila* (Owusu-Ansah et al., 2013, *Cell*). Moreover, there are several experimental limitations and interpretation issues that need to be addressed.

To draw a robust conclusion on how redox signaling influences autophagy with important consequences for longevity, the authors should examine whether overexpression of other antioxidant enzymes, such as glutathione peroxidase, has similar to catalase effects on lifespan.

Did the authors assess ROS levels in catalase overexpressing flies compared to controls?

The authors show that both females and males that overexpress catalase are resistant to oxidative stress; however, Keap1/Nrf2 signaling is not induced in these flies with age as occurs in their age-matched controls. The authors should define the mechanism behind the increased resistance upon catalase overexpression.

The study points to redox regulated Atg4 as a central player of lifespan regulation upon catalase overexpression in female flies. Induction of autophagy in catalase overexpressing flies has been shown by Western blot analysis. It would be nice to monitor autophagy *in vivo* by quantifying the number of Atg8-GFP punctae and also estimate autophagic flux upon catalase overexpression.

Furthermore, the authors should test whether mitophagy is induced in catalase overexpressing flies and define its role in lifespan extension under their experimental conditions.

Since mitochondria are thought to be the major source of intracellular ROS production, it is important to investigate whether mitochondrial targeted catalase overexpression has a specific role in lifespan regulation.

The authors fail to detect differences in TAG levels and glycogen storage between wild-type and catalase overexpressing females assayed at d7 and during a short period of starvation (as shown in Fig. 2 d and e, respectively). They should measure lipid content during aging to test whether there are differences in TAG levels between wild-type and catalase overexpressing flies later in life (after the switch back to *ad libitum* feeding) as is the case with intermittent fasting (Catterson et al., *Current Biology*, 2018). Such differences would contribute to lifespan extension in catalase overexpressing flies. Notably, a recent study has linked catalase overexpression with changes in metabolic profile (lipid profile and glucose levels) in a new "stress-less: leptin deficient mouse model (Amos et al, 2017, *BBA - Molecular Basis of Disease*).

The authors note that they performed their experiments in the white Dahomey (wDAH) genetic background, a "long-lived and outbred WT strain". I believe it is crucial that they show a similar (or even stronger) lifespan expression in an additional wild-type, not inherently long-lived strain.

The authors should experimentally show that the catalase they overexpress is functional and can manipulate H₂O₂ levels. Is this the peroxisomal catalase? Is it the only catalase homologue encoded by the *Drosophila* genome?

The gender-specific effects of catalase overexpression on lifespan are not adequately explained. Why do males fail to live longer? Does cat overexpression induce autophagy in male flies, similarly to their female siblings? If not, this could explain the observation that male flies do not live longer upon cat O/E. Furthermore, is the activity of the KEAP/NRF2 system also compromised in male flies similarly to their female siblings?

A main question is why are females so sensitive to acute starvation stress (Fig.2C). Starvation is normally associated with induction of autophagy. Why does a strain which already exhibits induced autophagy due to cat O/E so sensitive to starvation? I believe that the observation that the activity of KEAP/NRF2 system declines along aging upon cat overexpression (Fig.S3E) cannot convincingly explain why cat O/E females succumb to starvation while on the other hand have increased lifespan and healthspan. In other words, why is an otherwise long-lived and resistant to other stresses (like oxidative stress, Fig.1F-H) strain so sensitive to starvation stress specifically?

The authors provide compelling evidence that the oxidation status of numerous cellular proteins is influenced by cat

overexpression (Table S2). It is intriguing that mutation in a single residue in the key autophagy component Atg4A (Cys102) suffices to completely block the lifespan extension conferred by cat overexpression (figure 4C-D). Since the oxidation status of several mitochondrial and ribosomal proteins is affected by cat overexpression, and both mitochondrial metabolism and rate of protein synthesis affect lifespan, one can wonder whether the oxidation-reduction of such proteins is dispensable for lifespan extension upon cat overexpression. Moreover, I believe it would be helpful to annotate the proteins listed in table S2 according to their function to provide a gene ontology analysis.

Can the administration of H₂O₂ scavengers (such as sodium pyruvate or DMTU) mimic the beneficial effects of catalase overexpression in fly lifespan?

Tissue specific upregulation of catalase in muscles and ovaries might be informative about the lifespan extending mechanism and should be included in the relative analysis.

Lifespan analysis of male vs female Atg4 mutants and overexpressors could shed light on the females-specific effect of catalase overexpression.

Autophagy is measured as Atg8I or II / actin ratio. From the graph it is not clear if lipidated AtgII form is prominent in da-Gal4/UAS-cat flies. Alternatively, AtgII/AtgI ratio should be measured. Also, all controls should be included in the relative analysis and samples should have equal amounts of actin.

Researchers must prove that da-GS/UAS-Atg5RNAi and da-GS/UAS-cat+UAS-Atg5RNAi flies have reduced autophagy levels through western analysis or microscopy.

Minor:

In panel 1h the red dashed line denoting the survival of untreated females of the da-GAL4>UAS-cat genetic background is not evident in the graph.

In figure 1b, UAS-cat/+ flies should be included in the experiment.

Fecundity of da-Gal/UAS-cat should be tested to examine if redox activity increases longevity through reducing fecundity.

Reviewer #3

(Remarks to the Author)

In this manuscript the authors identify the role of autophagy in lifespan extension in response to increased expression of catalase. Previous studies of catalase overexpression failed to detect a significant effect on lifespan, however here the authors produced flies with substantially higher levels of the enzyme which resulted in an unexpected observation – the fly tissues appear to be characterised by a more oxidising environment which correlated with increased lifespan (in females only). This lifespan extension was cancelled by chemical or genetic suppression of autophagy and, more specifically, by mutating redox sensitive cysteine residue in Atg4 previously suggested to be important for the activation of autophagy by ROS in mammalian cells. Overall the manuscript is interesting, the data is of high quality and well presented. At the same time some key points have to be clarified before the manuscript can be recommended for publication.

1. The text of the manuscript would benefit from a clearer description of experimental rationale and data interpretation. For example, it would be good to explain from the first sentence what is the hypothesis that is being tested by catalase OE (one would assume that the expectation is that catalase will reduce H₂O₂ rather than affect “endogenous redox signalling”). The fact that this is not what was found by proteomics is not an issue as long as it is clearly explained (rather than having to interpret vague terms like “redox regulation” and “redox shift”). The diagram in Fig. 4e is also not helping in this regard: does it indicate that catalase blocks H₂O₂ or blocks H₂O₂ decrease?

2. Presumably the latter is correct as proteomics data suggests that catalase overexpression creates oxidising environment in the fly tissues. How (and whether) overexpression of antioxidant enzyme leads to increased levels of ROS is not clear, although it is explained by the perturbation of other antioxidant pathways (e.g. Nrf2). What is missing is an investigation whether ROS are increased in these flies and, if so, what is the source of these ROS (for example the authors can use available to them mitochondria-targeted hydrogen peroxide probe, Cochemé, et al, Cell Metab, 2011). If cellular (or specifically mitochondrial) ROS are not increased compared to age-matched controls (which would explain the lack of Nrf2 upregulation with age in catalase OE flies) perhaps the authors can provide an alternative explanation as to how increased oxidation of proteins is driven in this model. For example, would upregulation of Nrf2 or other antioxidants suppress the effect of catalase?

3. The authors conclude that catalase OE perturbs normal redox balance which increases oxidation of specific proteins, such as Atg4 on its regulatory cysteine (Cys102). At the same time the evidence for this selective oxidation of the regulatory but not catalytic cysteine is lacking. The authors can attempt IP of Atg4 from control and catalase OE flies to characterise its oxidation state. The Flag tag, although undetectable by blotting, may be sufficient for the enrichment of the protein for mass spectrometry analyses.

4. Since autophagy is proposed as a key mechanism underlying the effect of catalase OE, other markers of autophagy activity (e.g. Ref(2)p and ubiquitinated proteins) should be used to characterise its changes in addition to Atg8.

5. Minor points: Please indicate sex/strain in Fig.S1K for consistency.

Reviewer #4

(Remarks to the Author)

The Cochemé et al manuscript, "Enhancing autophagy by redox regulation extends lifespan in *Drosophila*," represents an attempt to further explore the potential roles of redox signaling in mediating longevity effects. Specifically, the authors establish that broad over-expression of catalase dampens down the age-related adaptive response mediated by NRF2, engenders a more oxidized state based on an increase in proteins associated with oxidized cysteine residues, and activates autophagy, resulting in sex-specific life span extension. Notably they present evidence that the presence of an oxidizable Cys in Atg4a is required for this life span extension. It is my view that these findings are sufficiently novel to merit a broad audience.

I have two moderate concerns and a few minor quibbles.

Concerns:

1. The authors clearly have the capacity to do redox proteomics and a strong addition to this would be to examine specifically the redox state of the Atg4a cys102 in the catalase over-expressor. I was hoping that their OxICAT analysis might have detected something, but I personally could not find Atg4 on the list in Table 2.
2. With regard to the Atg2 cys-ser mutant, the basis for the conclusion that the effects of catalase are mediated by this cysteine relies solely on a modest longevity effect. The inclusion of health span effects such as climbing activity would have provided a more compelling argument.

Quibbles:

1. It is stated in lines 37 and 100 that the catalase over-expressing flies are "acutely sensitive" to starvation stress. I imagined that "acutely" would translate into a dramatic effect, which indeed was not the case. Perhaps, surprisingly sensitive?
2. It is stated in line 54 that overexpressor flies were mildly delayed in eclosing. Were any sex-specific effects noted here?
3. With regard to the noted sex-specific differences, I was disappointed to see the rather early abandonment of the males in this study. For instance, it may have been informative to also test the catalase overexpressor males for up-regulation of autophagy (fig 2f), to see if there is a sex-specific disparity in this response. Similarly, is the oxidizing shift in the females also seen in the males in response to catalase overexpression?
4. Supplementary figure 2a would seem to argue that fecundity is reduced in the cat overexpressing animals. Please clarify.

Bill Orr

Version 1:

Reviewer comments:

Reviewer #1

(Remarks to the Author)

I recommend the manuscript to be accepted for publication

Reviewer #2

(Remarks to the Author)

In the revised manuscript, the authors have successfully addressed the majority of the reviewers' comments experimentally and by reviewing relevant literature. The new manuscript is improved and more solid, except for the autophagy section. While a role of Atg4-Cys102 oxidation in lifespan extension is supported, several points concerning results in autophagy deserve careful characterization, interpretation and revision.

The authors made use of LysoTracker Red and Cyto ID to corroborate western blot results for autophagy monitoring and autophagic flux estimation. There are several points of misinterpretation of LysoTracker Red/Cyto ID results.

1. In lines 150-151, the authors mention "To monitor autophagy status, we stained fly midguts with LysoTracker Red to label autophagolysosomes (Fig. 2f)." LysoTracker Red stains acidic compartments such as late endosomes and lysosomes. Autolysosomes could be a subpopulation of LysoTracker Red stained organelles. However, (increase in) LysoTracker Red puncta observed in Fig. 2f cannot be regarded as autolysosome labeling. Please review the description and the interpretation accordingly.

2. Supplementary Fig. 2c and lines 151-153:

- a. Why are autophagosomes (Cyto ID) not present upon autophagy induction (starvation)? Could the autophagosomal turnover be so fast that CytoID cannot capture autophagosomes under starvation conditions?
- b. Why do basal levels of autophagosomes (Cyto ID) not accumulate upon late-step autophagy inhibition (chloroquine)? Cyto ID stained organelles appear only colocalized with LysoTracker Red, probably signifying autolysosomes. However, autolysosome formation should be inhibited in this condition. Chloroquine treatment should block autophagosome-lysosome fusion. Therefore, autophagosomes should accumulate in distinct puncta (LysoTracker Red negative). How can these observations be explained?

3. LysoTracker Red puncta possibly coalesce in enormous-size structures (compared to the size of nuclei), a typical phenotype caused by lysosomotropic agent treatment (Sup. Fig. 2c). Such structures seem similar to catalase-

overexpressing flies in Fig. 2f, Fig. 4d, Sup. Fig. 2d and Sup. Fig. 4f, suggesting autophagic flux blockage in contrast to the authors' conclusion in lines 153-155. In addition, accumulation of both Atg8-I and Atg8-II in catalase flies (Fig. 4c) supports the above hypothesis. Furthermore, quantification of approximately 3 puncta per nucleus does not seem to agree with the images in Fig. 2f and 4d. What was exactly (the range of structures) quantified as a single punctum?

Overall, increase of LysoTracker Red puncta number reflects increased number of acidic compartments such as late endosomes and lysosomes in cells and does not necessarily signify autophagy induction. Cyto ID stain should at least indicate some non-acidic autophagosomal structures. Starvation should induce autophagy (shift in Atg8 lipidation in WB ratio and autophagosome formation in staining experiments). Fed, starved, chloroquine and starved/chloroquine conditions have to be compared for autophagic flux estimation in WB and staining experiments.

Regarding comment 8 of Reviewer 2:

Indeed, achieving lifespan extension in an already healthy long-lived strain is of great value. The authors can still validate (not repeat) their lifespan results in a healthy, not inherently long-lived wild type strain. What would a failure of extending lifespan mean, in that case?

Regarding comment 16 of Reviewer 2:

The Atg8-II/Atg8-I ratio is extremely informative and intrinsically normalized. It's of utmost significance to be measured and presented. Did samples run in the same or different gels/blots (since the lanes are in separate boxes in Fig. 2g and 4c)?

Reviewer #3

(Remarks to the Author)

I believe the revised manuscript has been substantially improved and can be recommended for publication.

Version 2:

Reviewer comments:

Reviewer #2

(Remarks to the Author)

The authors have now adequately responded to my points from the previous round of review. The revised manuscript is improved and I have no further comments.

Detailed response to reviewers' comments (NCOMMS-19-28345-T)

For clarity, comments from the reviewers are colour-coded *blue*, our responses are in *black*. Line numbers refer to numbering in the revised manuscript.

We thank the reviewers and the editor for their helpful comments in support of our study. We appreciate their patience and understanding for the delay in resubmitting our revised manuscript, due to pandemic-associated lab closures and challenges, as well as technically time-consuming efforts to develop targeted redox proteomics.

Reviewer #1:

Cocheme and coauthors have shown an importance of redox regulation in relation to longevity and stress resistance. Authors also showed that this mechanism works by affecting oxidation of specific residues in atg proteins. The work might be of great interest for scientific community but still some questions remained to make solid conclusions.

We thank Reviewer #1 for their positive overview of our study, and have addressed any outstanding questions below.

They are as follow:

1) I wonder why authors have chosen 5S-10Y diet for most experiments? Fruit fly require more sugar for successful survival and high protein diet shortens the lifespan. Taken into account that yeasts have about 45% of protein and 24% carbs than P:C ratio of experimental diet is about 1:2 that is rather to maximize reproduction than lifespan. Data from Lee et al. (2009) show that P:C 1:8 is more supportive for maximal lifespan. So, the presented data was generated at rather protein-enriched diet that is presumable toxic for flies.

The SYA diet has been extensively optimised for fly lifespan and fecundity in the Partridge laboratory (e.g. Bass *et al.* 2007, *J. Gerontol* - PMID 17921418). From these experiments, the 5%S,10%Y diet was selected as optimal, exhibiting long lifespan and high fecundity, but with both these parameters still responsive to DR (e.g. Figs. 2a and S2a). From our DR analysis (Fig. 2a), we show that the *da-GAL4>UAS-cat* over-expressors are long-lived at all yeast concentrations in the range 1-15%Y, where the sugar is maintained constant at 5%S. We find that increasing the % sugar (i.e. 10%S,10%Y and 20%S,10%Y) actually inhibits egg laying (e.g. Bass *et al.* 2007, *J Gerontol* - PMID 17921418; van Dam *et al.* 2020 *Cell Metab* - PMID 32197072).

2) I couldn't find an information whether genetic manipulations affected an amount of food consumption? It is also reasonable in terms of determination of resistance to oxidative stress under H₂O₂ or paraquat. Latter one strongly decreases food palatability. So, the question is did all flies consumed same amount of oxidants? Moreover, did consumption of hydrogen peroxide and paraquat induced oxidative stress???

Food consumption may indeed affect the dose of oxidant ingested. Therefore, in addition to diet-based oxidative stress, we also performed non-diet interventions: 1) paraquat injections (where a constant dose of paraquat is microinjected into the fly body via a glass capillary) (Fig. 1h), and 2) environmental hyperoxia (where flies are incubated in a 90% O₂ chamber on standard SYA food) (Figs. 1i and S1t). Catalase over-expressors were robustly resistant in both these non-diet stress assays. Furthermore, from our DR data (Fig. 2a), we show that the catalase flies are longer-lived at all yeast concentrations (i.e. they exhibit a vertical shift in the DR tent). If the lifespan extension was via DR from differences in food consumption, then we would expect to observe a horizontal shift, which is not the case. Therefore, differences in food consumption do not underlie the survival effects under both challenged and unchallenged conditions.

3) Authors have overexpressed catalase and showed it increased amount by WB. I wonder if they have measured enzymatic activity of catalase. WB may represent an increase in protein but it is not necessary that it is active. Antibodies can also bind inactive apoenzyme.

This is an extremely valid point. As suggested, in addition to assessing catalase over-expression by QPCR and western blotting, we now measure catalase enzyme activity biochemically, and show that catalase activity levels are significantly increased in the over-expressors. These new data are presented as Figs. S1d and S4c in the revised manuscript, and discussed in the text (lines 80-82 and 224-226). The methods section has been updated accordingly (lines 416-425).

4) As far as I know hydrogen peroxide itself cannot cause oxidation of cysteine residues. It may be due to its single electron reduction to generate hydroxyl radical that is very reactive. Moreover, residues can be oxidized by superoxide anion that is a substrate for SOD. I suggest to overexpress SOD and check some most affected parameters. What about other enzymes that detoxify hydrogen peroxide – peroxidases or transferases?

H₂O₂ is the main form of ROS implicated in redox signalling. It is widely described that H₂O₂ can react with cysteine residues in their reduced deprotonated form (i.e. thiolate R-S⁻) to give an oxidised sulfenic acid (R-SOH) and subsequently other oxidative post-translational modifications (e.g. Lennicke & Cochemé 2021 *Mol Cell* - PMID 34547234). Since the hydroxyl radical ([•]OH) is highly reactive in a diffusion-limited manner, this ROS is typically considered more in the context of indiscriminate oxidative damage, rather than physiological reversible redox regulation.

In addition to catalase, we have also overexpressed superoxide dismutase, either UAS-Sod1 or UAS-Sod2, ubiquitously using the da-GAL4 driver, and find that lifespan is not extended in females (Rebuttal Fig. 1). This is consistent with our working model, that the lifespan benefits occur through quenching of H₂O₂, which prevents endogenous redox signalling, resulting in a net oxidising shift in bulk cysteine thiols. This mimics a perceived state of state of starvation, ultimately leading to enhanced autophagy.

Rebuttal Figure 1 - Ubiquitous SOD1 or SOD2 overexpression does not increase survival in females. Experiments were set up with n~200 female flies per genotype. All lines were backcrossed into *w^{Dah}* background.

5) On the supplementary fig 1B authors showed that cat overexpression slows down the development. Might be shorter development a causative for extended lifespan showed on fig 1A?

We have shown that adult-specific over-expression using the inducible GeneSwitch system also extends lifespan (existing data Fig. 1b). These experiments bypass larval development, and therefore the lifespan extension is not due to developmental effects.

6) There is also no reasonable explanation why cat overexpression did not extend the lifespan in male flies. Is presented mechanism works only in females??

We indeed find that ubiquitous catalase overexpression does not extend lifespan (Fig. 1a) or induce starvation sensitivity in males (Fig. 2c). We believe that a recent study from our group sheds light on the sex-specificity of the catalase effect, by showing that basal autophagy levels are higher in male flies (Regan *et al.* 2022 *Nat Aging* - PMID 37118538). This suggests that females have a broader

'metabolic plasticity' and are more likely to benefit from interventions that enhance autophagy. We now include this study in our discussion as a reasonable explanation why males do not show lifespan extension upon catalase up-regulation (lines 259-263 of the revised manuscript).

7) Additionally, as shown on supplementary fig2A, fecundity was decreased in cat overexpressing flies that can be a reason for lifespan modulation in female flies.

Fecundity is indeed slightly decreased in the catalase over-expressing females (Fig. S2a). We attribute this to altered NOX (NADPH oxidase) signalling: NOX-derived H₂O₂ is required for oviduct muscle contraction necessary for egg laying (Ritsick *et al.* 2007, *Free Rad Biol Med* - PMID 17561091). Ubiquitous NOX-RNAi flies (i.e. decreased H₂O₂ production) exhibit an egg retention phenotype, which is phenocopied by the catalase over-expressors (i.e. increased H₂O₂ scavenging).

Altered fecundity does not necessarily result in a lifespan trade-off: e.g. rapamycin treatment still extends survival in sterile ovoD flies (Bjedov *et al.* 2010 *Cell Metab* - PMID 20074526). Indeed, catalase over-expression in the Atg4-Cys mutant background still exhibit decreased egg laying without extending lifespan. Therefore, the increased survival upon catalase up-regulation cannot be explained by altered fecundity. These new data are now shown in Fig. S4h of the revised manuscript and discussed in the text (lines 241-243).

8) I also wonder if diet supplementation with antioxidant containing thiol groups will give the same phenotype. Cysteine, glutathione or DTT can be tested.

We agree this is an interesting question. In fact, a recent study in *C. elegans* (Gusarov *et al.* 2021 *Nat Commun* - PMID 34267196) has shown that excess dietary supplementation with NAC (N-acetyl cysteine) accelerates ageing by inhibiting *skn-1*-mediated transcription, the worm orthologue of Nrf2. Therefore, manipulating redox state via dietary interventions can affect lifespan in worms, and would be worth exploring further in flies in future studies.

Reviewer #2 (Remarks to the Author):

In this manuscript, Cochemé and colleagues explored the role of redox signaling in the regulation of aging and metabolic health, using *Drosophila* as a model organism. More specifically, the authors show that overexpression of catalase, driven by ubiquitous promoters early in adulthood, is required to enhance healthspan and longevity in female flies. Notably, both female and male flies that overexpress catalase are resistant to multiple modes of oxidative stress. In an effort to gain insight into the mechanism underlying the longevity-promoting effects of catalase overexpression, the authors found that catalase over-expressing female flies exhibited a normal response to dietary restriction (DR). However, these flies displayed an increased starvation sensitivity compared to controls despite the fact that the levels of triacylglyceride (TAG) and glycogen storage did not differ significantly between control and catalase overexpressing flies. Furthermore, catalase overexpressing flies had increased levels of autophagy and their lifespan extension was abolished by knockdown of Atg5. Implementation of the oxidative isotope-coded affinity tags (OxiCAT) approach revealed that catalase overexpressing female flies displayed an oxidizing shift in cysteine redox state relative to controls during aging. This shift was dependent on the redox sensitive cysteine residue Cy102 in Atg4, the only cysteine peptidase in the autophagic pathway. The authors mutated Atg4a-C102S by CRISPR-Cas9 and found that this mutation is sufficient to suppress lifespan extension conferred by catalase overexpression. The study provides evidence suggesting that ubiquitous overexpression of catalase (an enzyme that reduces hydrogen peroxide to water) extends lifespan in female flies. This finding is surprising since it has been previously shown that ectopic expression of catalase in the mitochondrion has either neutral or negative effects on longevity in *Drosophila* (Mockett *et al.*, 2010, *Free Radical Biology & Medicine*; Sanz, 2016, *Biochimica et Biophysica Acta*). In line with these findings, it has been recently shown that catalase overexpression abrogates the extended lifespan associated with mild muscle mitochondrial perturbations in *Drosophila* (Owusu-Ansah *et al.*, 2013, *Cell*). Moreover, there are several experimental limitations and interpretation issues that need to be addressed.

We thank Reviewer #2 for their constructive comments, and have addressed their questions below.

1) To draw a robust conclusion on how redox signaling influences autophagy with important consequences for longevity, the authors should examine whether overexpression of other antioxidant enzymes, such as glutathione peroxidase, has similar to catalase effects on lifespan.

Please see our response to Point 4 from Reviewer #1, who also asked about other antioxidant enzymes. We have over-expressed Sod1 or Sod2 ubiquitously, and unlike catalase, did not observe a lifespan extension in females (Rebuttal Figure 1).

2) Did the authors assess ROS levels in catalase overexpressing flies compared to controls?

As suggested, we have now assessed ROS levels in the catalase flies. We used the redox-sensitive fluorescent dye CellROX to show that the levels of cellular ROS are decreased upon catalase up-regulation. In parallel, we measured levels of mitochondrial H₂O₂ using the ratiometric mass spectrometry probe MitoB (Cochemé *et al.* 2011, *Cell Metab* - PMID 21356523). Consistent with the over-expressed catalase not being mitochondria-targeted, we found no effect on mitochondrial H₂O₂ levels. These new data are now included as Figs. S3f,g,h of the revised manuscript, and discussed in the text (lines 196-201). The methods section has been updated accordingly (lines 501-516 and 525-531).

3) The authors show that both females and males that overexpress catalase are resistant to oxidative stress; however, Keap1/Nrf2 signaling is not induced in these flies with age as occurs in their age-matched controls. The authors should define the mechanism behind the increased resistance upon catalase overexpression.

The resistance to oxidative stress is directly linked to the antioxidant function of catalase, which detoxifies H₂O₂ and protects against acute oxidative damage. This explains why both female and male flies are resistant to different modes of oxidative stress (induced by PQ, exogenous H₂O₂ treatment, hyperoxia) upon catalase up-regulation. Whereas the lifespan benefits in females are a physiological response to alterations in redox signalling.

4) The study points to redox regulated Atg4 as a central player of lifespan regulation upon catalase overexpression in female flies. Induction of autophagy in catalase overexpressing flies has been shown by Western blot analysis. It would be nice to monitor autophagy in vivo by quantifying the number of Atg8-GFP punctae and also estimate autophagic flux upon catalase overexpression.

We fully agree that an independent measure of autophagy to complement the existing Atg8 western blot data would be valuable to strengthen our findings. Therefore, we have performed further confocal imaging using LysoTracker and CytolD staining to measure autophagy levels and flux respectively (as described in Lu *et al.* 2021 *eLife* - PMID 33988501), which fully corroborates our Atg8 analysis. These new data are presented as Fig. 2f, Figs. S2c-d, Fig. 4d, and Fig. S4f, and discussed in the text (lines 151-155 and 229-234). The methods section has been updated accordingly (lines 491-500).

5) Furthermore, the authors should test whether mitophagy is induced in catalase overexpressing flies and define its role in lifespan extension under their experimental conditions.

We thank the reviewer for this suggestion. We have now assessed mitophagy levels using a fluorescent genetic reporter, UAS-mito-QC (described in Lee *et al.* 2018 *J Cell Biol* - PMID 29500189), and find no difference in the catalase over-expressors. These data are presented as a new figure (Fig. S2e), and mentioned in the revised manuscript (lines 155-157).

6) Since mitochondria are thought to be the major source of intracellular ROS production, it is important to investigate whether mitochondrial targeted catalase overexpression has a specific role in lifespan regulation.

As suggested, we have now performed additional lifespan experiments using a published UAS-mito-cat transgene (generated in Radyuk *et al.* 2010 *Free Rad Biol Med* - PMID 20869434). Under our experimental conditions (genetic background and diet), we find that ubiquitous overexpression of mitochondria-targeted catalase (da-GAL4>UAS-mito-cat) does not extend lifespan in females.

These new data are presented in Figs. S1e,f of the revised manuscript, and mentioned in the text (lines 85-89). Our observation is consistent with a previous report in the literature (Scialo *et al.* 2016 *Cell Metab* - PMID 27076081), which also does not find lifespan extension, in fact they observe that the da-GAL4>UAS-mito-cat flies are mildly short-lived. We now refer to this study in the text (line 87).

7) The authors fail to detect differences in TAG levels and glycogen storage between wild-type and catalase overexpressing females assayed at d7 and during a short period of starvation (as shown in Fig. 2 d and e, respectively). They should measure lipid content during aging to test whether there are differences in TAG levels between wild-type and catalase overexpressing flies later in life (after the switch back to ad libitum feeding) as is the case with intermittent fasting (Catterson *et al.*, *Current Biology*, 2018). Such differences would contribute to lifespan extension in catalase overexpressing flies. Notably, a recent study has linked catalase overexpression with changes in metabolic profile (lipid profile and glucose levels) in a new “stress-less: leptin deficient mouse model (Amos *et al.*, 2017, *BBA - Molecular Basis of Disease*).

As suggested, we have complemented our existing d7 TAG data (Fig. 2d), with additional TAG analysis performed at d28 of adulthood. We find similar TAG levels at d28 in control (UAS-cat/+ and da-GAL4/+) and catalase up-regulated flies (da-GAL4>UAS-cat), under basal conditions as well as in response to a starvation time course. These new data are included as Fig. S2b in the revised manuscript, and discussed in the text (lines 140-144).

8) The authors note that they performed their experiments in the white Dahomey (wDAH) genetic background, a “long-lived and outbred WT strain”. I believe it is crucial that they show a similar (or even stronger) lifespan expression in an additional wild-type, not inherently long-lived strain.

Genetic background is an important consideration in the ageing field (Partridge & Gems 2007 *Nature* - PMID 17994065). This study was performed in the outbred *w^{Dah}* wild-type and all lines were fully back-crossed into this background. Achieving lifespan extension in an already healthy long-lived strain is a valuable outcome. We feel that repeating the experiments in another not inherently long-lived strain would not be informative, due to the potential of rescuing an inherent sickness.

9) The authors should experimentally show that the catalase they overexpress is functional and can manipulate H₂O₂ levels. Is this the peroxisomal catalase? Is it the only catalase homologue encoded by the *Drosophila* genome?

As raised by Reviewer #1 Point 3, we have now assessed catalase enzyme activity and shown that the catalase is functional. Please see our detailed response above. We have also included ROS measurements using fluorescent CellROX staining to show decreased cellular ROS in catalase over-expressors (new Figs. S3f,g), and the MitoB probe to show unchanged levels of mitochondrial H₂O₂ (new Fig. S3h).

In addition to conventional intracellular catalase, *Drosophila* also encodes an extracellular immune-regulated catalase (IRC) which is involved in host protection against pathogen infection (e.g. Ha *et al.* 2005, *Dev. Cell* - PMID 15621536). In mRNA expression profiling, we did not detect any changes to IRC levels between control (UAS-cat/+) and catalase over-expressor (da-GAL4>UAS-cat) females (adjusted p value=0.497).

10) The gender-specific effects of catalase overexpression on lifespan are not adequately explained. Why do males fail to live longer? Does cat overexpression induce autophagy in male flies, similarly to their female siblings? If not, this could explain the observation that male flies do not live longer upon cat O/E. Furthermore, is the activity of the KEAP/NRF2 system also compromised in male flies similarly to their female siblings?

The sex-specificity of the catalase-induced lifespan effect was also raised by Reviewer #1 in their Point 6. Please see our detailed response above.

11) A main question is why are females so sensitive to acute starvation stress (Fig.2C). Starvation is normally associated with induction of autophagy. Why does a strain which already exhibits induced autophagy due to cat O/E so sensitive to starvation? I believe that the observation that the activity of KEAP/NRF2 system declines along aging upon cat overexpression (Fig.S3E) cannot convincingly

explain why cat O/E females succumb to starvation while on the other hand have increased lifespan and healthspan. In other words, why is an otherwise long-lived and resistant to other stresses (like oxidative stress, Fig.1F-H) strain so sensitive to starvation stress specifically?

In our model, the resistance to oxidative stress is due to the direct antioxidant properties of catalase, whereas the starvation response is downstream of altered redox signalling due to quenching of second messenger H₂O₂ by catalase. Catalase up-regulation leads to an oxidising thiol shift, which mimics an internal state of starvation, as described in our previous work (Menger *et al.* 2015 *Cell Rep* - PMID 26095360). Therefore, we propose that the catalase up-regulated flies are already in 'starvation response mode', which leads to lifespan benefits under fed conditions through redox-dependent autophagy up-regulation. From other recent work in our group (Bjedov *et al.* 2020 *PLOS Genet* - PMID 33253201), we show that levels of autophagy are carefully fine-tuned, and that excess autophagy activation can lead to starvation sensitivity *in vivo*. This could explain their sensitivity under subsequent starvation conditions.

12) The authors provide compelling evidence that the oxidation status of numerous cellular proteins is influenced by cat overexpression (Table S2). It is intriguing that mutation in a single residue in the key autophagy component Atg4A (Cys102) suffices to completely block the lifespan extension conferred by cat overexpression (figure 4C-D). Since the oxidation status of several mitochondrial and ribosomal proteins is affected by cat overexpression, and both mitochondrial metabolism and rate of protein synthesis affect lifespan, one can wonder whether the oxidation-reduction of such proteins is dispensable for lifespan extension upon cat overexpression. Moreover, I believe it would be helpful to annotate the proteins listed in table S2 according to their function to provide a gene ontology analysis.

As suggested, in addition to the list of oxidised Cys residues (Supplementary Table 2), we now provide functional information for the proteins detected in our OxICAT dataset, including GO molecular function and KEGG pathway analysis (Supplementary Table 3 in the revised manuscript). In our study, we focus on Atg4-Cys102 and provide detailed mechanistic follow-up using a CRISPR knock-in mutant. Other oxidative PTMs would need to be similarly studied in detail to make conclusions on their biological impact *in vivo*.

13) Can the administration of H₂O₂ scavengers (such as sodium pyruvate or DMTU) mimic the beneficial effects of catalase overexpression in fly lifespan?

Reviewer #1 Point 8 also asked about dietary supplementation with antioxidants. Please see our detailed response above.

14) Tissue specific upregulation of catalase in muscles and ovaries might be informative about the lifespan extending mechanism and should be included in the relative analysis.

In addition to the tissue-specific drivers already tested (tubules, IPC, pan-neural, gut and fat body - existing Figs. S1n,o,p,q,r), as suggested we have now performed an additional lifespan with a pan-muscular driver (MHC-GS, under the myosin heavy chain promoter). Muscle-specific catalase overexpression did not extend lifespan, and in fact MHC-GS>UAS-cat +RU flies appear to be slightly short-lived. Therefore, muscle effects cannot explain the lifespan benefits upon catalase up-regulation. These new data are now incorporated as Fig. S1s in the revised manuscript. We have already uncoupled the catalase lifespan effect from fecundity, therefore we did not explore ovary-specific expression. Please see our detailed response above to Reviewer #1 Point 7.

15) Lifespan analysis of male vs female Atg4 mutants and overexpressors could shed light on the females-specific effect of catalase overexpression.

As catalase up-regulation did not extend lifespan in males, we did not further explore Atg4-C102S mutants in males. We believe that the reason for the sex-specific effects is due to higher basal autophagy levels in males. Please see our detailed response above to Reviewer #1 Point 6.

16) Autophagy is measured as Atg8I or II / actin ratio. From the graph it is not clear if lipidated AtgII form is prominent in da-Gal4/UAS-cat flies. Alternatively, AtgII/AtgI ratio should be measured. Also, all controls should be included in the relative analysis and samples should have equal amounts of

actin. Researchers must prove that da-GS/UAS-Atg5RNAi and da-GS/UAS-cat+UAS-Atg5RNAi flies have reduced autophagy levels through western analysis or microscopy.

Our Atg8 WB analysis is the summary of n=6 replicates for Fig. 2g and n=2 replicates for Fig. 4c. The samples consisted of dissected abdomens, which occasionally introduced technical variability in total protein, hence why the Atg8 signal was normalised to actin as a loading control. While we acknowledge that including all the control lines within each experiment would be ideal, for practical purposes in some experiments, we used the UAS-cat/+ flies as the representative control. However, from our extensive prior characterisation (QPCR, WB, survival assays - Figs. 1 and S1) and normalisation of genetic background by backcrossing into w^{Dah} , we have shown that the UAS-cat transgene is not leaky (i.e. expressed in the absence of GAL4) and behaves similarly to the w^{Dah} WT and the driver-only flies (da-GAL4/+), therefore we are confident that our findings are robust.

The UAS-Atg5-RNAi line (originally published in Scott *et al.* 2004, *Dev Cell* - PMID 15296714) has been widely used by the field (>20 references according to FlyBase). We have already validated that this Atg5-RNAi decreases autophagy by Atg8 WB under our experimental conditions (backcrossed into w^{Dah} background, on an SYA diet) in another recent study from our group (Bjedov *et al.* 2020 *PLOS Genet* - PMID 33253201). Here, we have confirmed by QPCR that the same level of Atg5 RNAi knockdown is achieved in a catalase background (Fig. S2g). For clarity, we now refer to this validation study more explicitly in the main text (lines 168-169 of the revised manuscript).

Minor:

17) In panel 1h the red dashed line denoting the survival of untreated females of the da-GAL4>UAS-cat genetic background is not evident in the graph.

We apologise that the da-GAL4>UAS-cat line was not visible (there are no deaths in the uninjected flies, therefore this line overlaps with the UAS-cat/+ control). We have adjusted the plotting of Fig. 1h to make this clearer.

18) In figure 1b, UAS-cat/+ flies should be included in the experiment.

Similarly to the point raised in #16, from our QPCR data (Fig. S1a), we know that UAS-cat line is not leaky (i.e. it is not expressed in the absence of GAL4). We included the driver-only line (da-GS/+) ±RU, to show that the drug does not affect survival in a control line. Furthermore, the da-GS>UAS-cat -RU condition overlaps with these other controls, confirming that the da-GS driver is not leaky (i.e. it does not induce expression of the catalase transgene in the absence of RU drug).

19) Fecundity of da-Gal/UAS-cat should be tested to examine if redox activity increases longevity through reducing fecundity.

Reviewer #1 Point 7 also asked a question regarding fecundity. Please refer above to our combined response to this question.

Reviewer #3 (Remarks to the Author):

In this manuscript the authors identify the role of autophagy in lifespan extension in response to increased expression of catalase. Previous studies of catalase overexpression failed to detect a significant effect on lifespan, however here the authors produced flies with substantially higher levels of the enzyme which resulted in an unexpected observation – the fly tissues appear to be characterised by a more oxidising environment which correlated with increased lifespan (in females only). This lifespan extension was cancelled by chemical or genetic suppression of autophagy and, more specifically, by mutating redox sensitive cysteine residue in Atg4 previously suggested to be important for the activation of autophagy by ROS in mammalian cells. Overall the manuscript is interesting, the data is of high quality and well presented. At the same time some key points have to be clarified before the manuscript can be recommended for publication.

We thank Reviewer #3 for their favourable opinion regarding the interest, quality and presentation of our study. We have clarified their comments raised below:

1. The text of the manuscript would benefit from a clearer description of experimental rationale and data interpretation. For example, it would be good to explain from the first sentence what is the hypothesis that is being tested by catalase OE (one would assume that the expectation is that catalase will reduce H₂O₂ rather than affect “endogenous redox signalling”). The fact that this is not what was found by proteomics is not an issue as long as it is clearly explained (rather than having to interpret vague terms like “redox regulation” and “redox shift”). The diagram in Fig. 4e is also not helping in this regard: does it indicate that catalase blocks H₂O₂ or blocks H₂O₂ decrease?

Expanding the manuscript introduction has allowed us to address these points and to clarify the experimental rationale, as suggested. We also fully agree that the directionality of the original graphical summary (Fig. 4e in the initial submission) was unclear. We have now modified and improved the scheme to make this less ambiguous, which is now presented as Fig. 4g in the revised submission.

2. Presumably the latter is correct as proteomics data suggests that catalase overexpression creates oxidising environment in the fly tissues. How (and whether) overexpression of antioxidant enzyme leads to increased levels of ROS is not clear, although it is explained by the perturbation of other antioxidant pathways (e.g. Nrf2). What is missing is an investigation whether ROS are increased in these flies and, if so, what is the source of these ROS (for example the authors can use available to them mitochondria-targeted hydrogen peroxide probe, Cochemé, et al, Cell Metab, 2011). If cellular (or specifically mitochondrial) ROS are not increased compared to age-matched controls (which would explain the lack of Nrf2 upregulation with age in catalase OE flies) perhaps the authors can provide an alternative explanation as to how increased oxidation of proteins is driven in this model. For example, would upregulation of Nrf2 or other antioxidants suppress the effect of catalase?

Please see our response above to Point 2 from Reviewer #2, who also requested ROS measurements. The catalase flies show decreased levels of cellular ROS, assessed using the CellROX dye (Figs. S3f,g), but no change in mitochondrial H₂O₂ based on MitoB analysis (Fig. S3h). Our model is that by quenching H₂O₂, the physiological function of H₂O₂ as a second messenger in redox signalling is prevented in the catalase over-expressors, and therefore other redox-responsive pathways are not induced with age, for instance Nrf2. To explore this further, we have additionally measured total glutathione levels, whose synthesis is downstream of Nrf2. Total GSH was decreased in the catalase flies, which is compatible with the oxidised thiol shift, since GSH is an important redox cofactor. These new data are now presented as Fig. 3f of the revised manuscript, and discussed in the text (lines 211-213). The methods section has been updated accordingly (lines 427-437).

3. The authors conclude that catalase OE perturbs normal redox balance which increases oxidation of specific proteins, such as Atg4 on its regulatory cysteine (Cys102). At the same time the evidence for this selective oxidation of the regulatory but not catalytic cysteine is lacking. The authors can attempt IP of Atg4 from control and catalase OE flies to characterise its oxidation state. The Flag tag, although undetectable by blotting, may be sufficient for the enrichment of the protein for mass spectrometry analyses.

We agree with Review #3, and spent considerable time, effort and resources attempting to address this point. Our initial OxICAT proteomic analysis was unbiased, and while valuable in providing a snapshot of global thiol redox state, the proteins detected only represent ~1% of possible cysteine residues in the fly proteome. Furthermore, detected peptides mainly come from abundant metabolic enzymes or structural proteins, rather than less abundant regulatory proteins (discussed in Menger *et al.* 2015 *Cell Rep* - PMID 26095360). Unsurprisingly, Atg4a was not amongst these. As suggested, we first tried to IP Atg4a using the Flag tag, but this was not successful (we tested multiple Flag antibodies and incubation conditions). Since no antibodies against *Drosophila* Atg4a are commercially available, we then tested several antibodies against mammalian Atg4a (x3 different antibodies; either polyclonal or with epitope regions of high predicted conservation) in order to perform IPs, but none of these cross-reacted with fly Atg4a. Next, we generated our own custom antibodies for *Drosophila* Atg4a (raised against x2 different peptides, using a commercial company with whom we have successfully worked previously), but unfortunately these did not work by WB or IP.

Since IP-enrichment was not possible, we therefore pursued an alternative strategy and performed pH-based fractionation of total fly protein extracts, to identify the fraction containing Atg4a. However, while other peptides from Atg4a were clearly detected, showing that our enrichment was successful, the specific tryptic peptide containing Cys102 was problematic. Furthermore, *in silico* digests with a range of other proteolytic enzymes did not improve peptide cleavage yield. Our inability to observe this specific Atg4a peptide is consistent with other large-scale redox proteomic studies, particularly the OxiMouse dataset, which uses enhanced specific enrichment of Cys-containing peptides and covered ~34,000 unique Cys residues (Xiao *et al.* 2020, *Cell* PMID 32109415). Even then, the equivalent orthologue peptide (Cys81 in mouse) was not present, suggesting that Cys102 is not amenable to detection by mass spectrometry. A likely explanation is that Cys102 is the N-terminal residue of the peptide when Atg4a is trypsinised. The N-terminal amine of Cys102 may undergo modifications in a reaction catalysed by its thiol (e.g. James *et al.* 2022, *Cell Chem. Biol.* PMID 35868236, see also the process of Native Chemical Ligation), which results in an unpredictable parent ion mass.

While the reviewer's suggestion is a good one and we would like to show the modification, we don't think it's possible for the above technical reasons. Nevertheless, the strong bulk oxidising shifts detected by our OxiCAT redox proteomic analysis in response to both fasting and catalase up-regulation, combined with *in vivo* validation using the specific Atg4a-Cys102Ser genetic knock-in mutant support our model.

4. Since autophagy is proposed as a key mechanism underlying the effect of catalase OE, other markers of autophagy activity (e.g. Ref(2)p and ubiquitinated proteins) should be used to characterise its changes in addition to Atg8.

We fully agree with this suggestion, which was also raised by Reviewer #2, Point 4, and now provide new confocal microscopy imaging to complement our existing WB data. Please see above for our combined response.

5. Minor points: Please indicate sex/strain in Fig.S1K for consistency.

We apologise for this oversight, and have now included this information (Fig. S1p in the revised manuscript).

Reviewer #4 (Bill Orr):

The Cochemé *et al* manuscript, "Enhancing autophagy by redox regulation extends lifespan in *Drosophila*," represents an attempt to further explore the potential roles of redox signaling in mediating longevity effects. Specifically, the authors establish that broad over-expression of catalase dampens down the age-related adaptive response mediated by NRF2, engenders a more oxidized state based on an increase in proteins associated with oxidized cysteine residues, and activates autophagy, resulting in sex-specific life span extension. Notably they present evidence that the presence of an oxidizable Cys in Atg4a is required for this life span extension. It is my view that these findings are sufficiently novel to merit a broad audience. I have two moderate concerns and a few minor quibbles.

We thank Reviewer #4 for recognising the novelty of our study. We have addressed their remaining comments below:

Concerns:

1. The authors clearly have the capacity to do redox proteomics and a strong addition to this would be to examine specifically the redox state of the Atg4a cys102 in the catalase over-expressor. I was hoping that their OxiCAT analysis might have detected something, but I personally could not find Atg4 on the list in Table 2.

We fully agree with Reviewer #4, and this point was also raised by Reviewer #3. Atg4a was not amongst the proteins detected in the unbiased OxiCAT analysis. Therefore, we spent considerable time and resources attempting to address this point and perform targeted redox proteomics of

Cys102 in Atg4a. Please see our detailed response above to Reviewer #3 Point 3, detailing our efforts and the technical limitations.

2. With regard to the Atg2 cys-ser mutant, the basis for the conclusion that the effects of catalase are mediated by this cysteine relies solely on a modest longevity effect. The inclusion of health span effects such as climbing activity would have provided a more compelling argument.

As suggested, we have performed climbing activity assays to assess healthspan of the flies. These new data are now shown in Fig. S4g and discussed in the text (lines 238-241 of the revised manuscript).

Quibbles:

1. It is stated in lines 37 and 100 that the catalase over-expressing flies are “acutely sensitive” to starvation stress. I imagined that “acutely” would translate into a dramatic effect, which indeed was not the case. Perhaps, surprisingly sensitive?

We appreciate and agree with the reviewer’s point. We have now rephrased this and toned down our description by removing ‘acutely’ from the text (lines 29, 64, and 139 of the revised manuscript).

2. It is stated in line 54 that overexpressor flies were mildly delayed in eclosing. Were any sex-specific effects noted here?

The sex of the eclosing flies was recorded during the assay (Fig. S1g in the revised manuscript), so we can replot these data separately for females and males (Rebuttal Figure 2). As we did not observe a sex-specific effect for eclosion (i.e. both females and males were similarly mildly delayed) and also we showed that the lifespan extension upon catalase up-regulation was not due to developmental effects, we opted not to include this breakdown in the resubmission.

Rebuttal Figure 2 - Development time course plotted separately by sex. Constitutive ubiquitous catalase over-expression (da-GAL4>UAS-cat) causes a mild developmental delay in both females and males relative to the UAS-cat/+ and da-GAL4/+ controls. Synchronised L1 larvae (n=500 per genotype) were transferred to SYA vials, and the subsequent eclosion time of adults was recorded.

3. With regard to the noted sex-specific differences, I was disappointed to see the rather early abandonment of the males in this study. For instance, it may have been informative to also test the catalase overexpressor males for up-regulation of autophagy (fig 2f), to see if there is a sex-specific disparity in this response. Similarly, is the oxidizing shift in the females also seen in the males in response to catalase overexpression?

The question of sex-specificity was also raised by Reviewers #1 and #2. Please see our combined response on this topic under Reviewer #1, Point 6.

4. Supplementary figure 2a would seem to argue that fecundity is reduced in the cat overexpressing animals. Please clarify.

Fecundity was also raised by Reviewers #1 and #2. Please refer to our combined response above under Reviewer #1, Point 7.

Detailed response to reviewers' comments (NCOMMS-19-28345-A) - 2nd revision

We thank Reviewers #1 and #3 for their support and recommending our revised manuscript for publication. Below, we address the residual comments from Reviewer #2.

Reviewer #2 (Remarks to the Author):

In the revised manuscript, the authors have successfully addressed the majority of the reviewers' comments experimentally and by reviewing relevant literature. The new manuscript is improved and more solid, except for the autophagy section. While a role of Atg4-Cys102 oxidation in lifespan extension is supported, several points concerning results in autophagy deserve careful characterization, interpretation and revision.

We thank Reviewer #2 for recognising the improvements to our manuscript and our efforts in addressing the reviewers' comments overall. Please find below our responses to the remaining points, particularly regarding the autophagy assays. We appreciate this feedback, which we have incorporated into our 2nd revised manuscript and has helped to further strengthen our study.

The authors made use of LysoTracker Red and Cyto ID to corroborate western blot results for autophagy monitoring and autophagic flux estimation. There are several points of misinterpretation of LysoTracker Red/Cyto ID results.

1. In lines 150-151, the authors mention "To monitor autophagy status, we stained fly midguts with LysoTracker Red to label autophagolysosomes (Fig. 2f)." LysoTracker Red stains acidic compartments such as late endosomes and lysosomes. Autolysosomes could be a subpopulation of LysoTracker Red stained organelles. However, (increase in) LysoTracker Red puncta observed in Fig. 2f cannot be regarded as autolysosome labeling. Please review the description and the interpretation accordingly.

We agree and have rephrased the text as follows for clarification: "To monitor autophagy status, we stained fly midguts with LysoTracker Red, which labels acidic compartments such as late endosomes and lysosomes, including autophagolysosomes" (lines 150-152 of the re-revised manuscript).

2. Supplementary Fig. 2c and lines 151-153:

a. Why are autophagosomes (Cyto ID) not present upon autophagy induction (starvation)? Could the autophagosomal turnover be so fast that CytoID cannot capture autophagosomes under starvation conditions?

b. Why do basal levels of autophagosomes (Cyto ID) not accumulate upon late-step autophagy inhibition (chloroquine)? Cyto ID stained organelles appear only colocalized with LysoTracker Red, probably signifying autolysosomes. However, autolysosome formation should be inhibited in this condition. Chloroquine treatment should block autophagosome-lysosome fusion. Therefore, autophagosomes should accumulate in distinct puncta (LysoTracker Red negative). How can these observations be explained?

In our hands, we did not observe LysoTracker Red negative/CytoID Green positive punctae under the conditions tested. Indeed, autophagy may be progressing too quickly to allow the accumulation of autophagosomes, which would likely explain these observations.

We have now repeated our previous experiments using a new CytoID Autophagy Detection Kit 2.0 from Enzo. This CytoID 2.0 reagent is reported to be brighter and more photostable for autophagic vesicles. As a positive control suggested by the manufacturer, we have used rapamycin treatment to stimulate autophagy (in our case, by feeding flies this drug in their diet), which is confirmed by increased LysoTracker Red staining, but low CytoID 2.0 since autophagic flux is not blocked. However, when we co-treat flies by feeding rapamycin then additionally the inhibitor chloroquine, we now detect strong CytoID 2.0 staining, consistent with autophagy being both induced and blocked. These new data with CytoID 2.0 are presented as a new Supplementary Fig. 2d, replacing the previous version. We have updated the results, figure legend and methods sections accordingly.

Our new data with *in vivo* imaging of dissected guts (Supplementary Fig. 2d) is consistent with the observed pattern of response for cell culture published by the manufacturer in their CytolD 2.0 kit manual (screenshot of their graph included above - <https://www.enzo.com/product/cyto-id-autophagy-detection-kit-2-0/>), showing CytolD Green 2.0 staining only when autophagy is both stimulated by rapamycin and inhibited by chloroquine.

3. LysoTracker Red puncta possibly coalesce in enormous-size structures (compared to the size of nuclei), a typical phenotype caused by lysosomotropic agent treatment (Sup. Fig. 2c). Such structures seem similar to catalase-overexpressing flies in Fig. 2f, Fig. 4d, Sup. Fig. 2d and Sup. Fig. 4f, suggesting autophagic flux blockage in contrast to the authors' conclusion in lines 153-155. In addition, accumulation of both Atg8-I and Atg8-II in catalase flies (Fig. 4c) supports the above hypothesis. Furthermore, quantification of approximately 3 puncta per nucleus does not seem to agree with the images in Fig. 2f and 4d. What was exactly (the range of structures) quantified as a single punctum?

Overall, increase of LysoTracker Red puncta number reflects increased number of acidic compartments such as late endosomes and lysosomes in cells and does not necessarily signify autophagy induction. Cyto ID stain should at least indicate some non-acidic autophagosomal structures. Starvation should induce autophagy (shift in Atg8 lipidation in WB ratio and autophagosome formation in staining experiments). Fed, starved, chloroquine and starved/chloroquine conditions have to be compared for autophagic flux estimation in WB and staining experiments.

Regarding the number of puncta quantified, we fully agree with Reviewer #2 and apologise for this oversight. We have now re-analysed the confocal images using CellProfiler software and obtain mean values in the order of ~2 and ~25 punctae per cell for control and catalase over-expressor conditions respectively, which is more consistent with the representative images. We have amended the methods section accordingly (lines 503-505 in the re-revised manuscript, and new reference #72). These new graphs are presented as Figs. 2f and 4d, replacing the previous versions. While the absolute numbers are updated by this re-analysis, the relative comparisons between genotypes remain the same, and therefore our conclusions are not affected.

Following the above comment (Point 2b), we have now repeated our CytolD staining on the catalase control and over-expressor flies using the new CytolD 2.0 Reagent, and importantly have included the additional condition of chloroquine treatment. CytolD 2.0 staining is observed in the catalase up-regulated flies only upon chloroquine inhibition. This supports our interpretation that autophagy is induced in the catalase over-expressors but not blocked. If flux was affected then we would already see CytolD 2.0 accumulation in the catalase over-expressors even in the absence of chloroquine, which is not the case. These new experiments are presented as Supplementary Fig. 2e, replacing the previous version (Supplementary Fig. 2d in the old manuscript). We have updated the results section and figure legend accordingly.

Regarding comment 8 of Reviewer 2:

Indeed, achieving lifespan extension in an already healthy long-lived strain is of great value. The authors can still validate (not repeat) their lifespan results in a healthy, not inherently long-lived wild type strain. What would a failure of extending lifespan mean, in that case?

As previously stated, achieving lifespan extension in an already healthy long-lived wild-type is an important finding, and we feel that repeating these experiments in another not long-lived strain would not be informative. Strains can be short-lived for a variety of reasons, and rescuing an inherent sickness would not be meaningful without additional detailed phenotyping. Such investigations could be considered more valuable in the context of age-related disease models and their pathophysiology, however are beyond the scope of our current study.

Regarding comment 16 of Reviewer 2:

The Atg8-II/Atg8-I ratio is extremely informative and intrinsically normalized. It's of utmost significance to be measured and presented. Did samples run in the same or different gels/blots (since the lanes are in separate boxes in Fig. 2g and 4c)?

As requested, we have plotted the Atg8-II/Atg8-I ratio for the Western blot in Fig. 2g, now included as a new Supplemental Fig. 2g. Since Atg8-II is the lipidated form that localises to autophagosomes, we primarily focused our interpretation of induced autophagy from the Atg8-II/actin ratio based on the published "Guidelines for the use and interpretation of assays for monitoring autophagy" (Klionsky *et al.* 2021, PMID: 33634751), which states: 'Levels of LC3-II should be compared not to LC3-I [...], but ideally to more than one "housekeeping" protein such as actin.' Levels of Atg8-I vary according to the organism, tissue and biological condition, but can still be informative, hence why we have also included its quantitation in our analysis. The representative band images shown in Figures 2g and 4c were from the same experiment - the full blot with all lanes is provided as Supplemental Figure 4e. Critically, these guidelines highlight the importance of assessing autophagy by an independent orthologous assay, which we have done by performing confocal imaging with LysoTracker Red and CytolD Green staining. Overall, the enhanced LysoTracker Red staining, the lack of CytolD staining under basal conditions, together with the increased levels of lipidated Atg8-II, support our interpretation that autophagy is induced but flux is not blocked in the catalase flies.